# Microbially mediated mechanisms underlie soil carbon accrual by conservation agriculture under decade-long warming

Jing Tian [1,14] ✉, Jennifer A. J. Dungait [2,3,14], Ruixing Hou [4,14], Ye Deng [5,14], Iain P. Hartley [2], Yunfeng Yang [6], Yakov Kuzyakov [7], Fusuo Zhang [1] ✉, M. Francesca Cotrufo [8] ✉ & Jizhong Zhou [9,10,11,12,13] ✉

Increasing soil organic carbon (SOC) in croplands by switching from conventional to conservation management may be hampered by stimulated microbial decomposition under warming. Here, we test the interactive effects of agricultural management and warming on SOC persistence and underlying microbial mechanisms in a decade-long controlled experiment on a wheat-maize cropping system. Warming increased SOC content and accelerated fungal community temporal turnover under conservation agriculture (no tillage, chopped crop residue), but not under conventional agriculture (annual tillage, crop residue removed). Microbial carbon use efficiency (CUE) and growth increased linearly over time, with stronger positive warming effects after 5 years under conservation agriculture. According to structural equation models, these increases arose from greater carbon inputs from the crops, which indirectly controlled microbial CUE via changes in fungal communities. As a result, fungal necromass increased from 28 to 53%, emerging as the strongest predictor of SOC content. Collectively, our results demonstrate how management and climatic factors can interact to alter microbial community composition, physiology and functions and, in turn, SOC formation and accrual in croplands.

Soil organic carbon (SOC) stands as a property of ecosystem, offering a wide range of benefits to both human society and the natural environment, including global climate regulation[1,2]. However, the state of agriculture, which covers 38% of the Earth's land surface, is a matter of concern. Large swathes of agricultural land have suffered from moderate to severe degradation due to inappropriate management practices[3], which has reduced one-half to two-thirds of total SOC content compared with natural or uncultivated soils[4]. Addressing this

[1]State Key Laboratory of Nutrient Use and Management, College of Resources and Environmental Sciences, China Agricultural University, 100193 Beijing, PR China. [2]Geography, Faculty of Environment, Science and Economy, University of Exeter, Rennes Drive, Exeter EX4 4RJ, UK. [3]Carbon Management Centre, SRUC-Scotland's Rural College, Edinburgh EH9 3JG, UK. [4]Key Laboratory of Ecosystem Network Observation and Modeling, Institute of Geographic Sciences and Natural Resources Research, Chinese Academy of Sciences (CAS), 100101 Beijing, PR China. [5]CAS Key Laboratory for Environmental Biotechnology, Research Center for Eco-Environmental Sciences, Chinese Academy of Sciences, 100085 Beijing, PR China. [6]State Key Joint Laboratory of Environment Simulation and Pollution Control, School of Environment, Tsinghua University, Beijing, PR China. [7]Department of Soil Science of Temperate Ecosystems, University of Göttingen, 37077 Göttingen, Germany. [8]Department of Soil and Crop Science, Colorado State University, Fort Collins, CO, USA. [9]Institute for Environmental Genomics, University of Oklahoma, Norman, OK, USA. [10]School of Biological Sciences, University of Oklahoma, Norman, OK, USA. [11]School of Civil Engineering and Environmental Sciences, University of Oklahoma, Norman, OK, USA. [12]School of Computer Science, University of Oklahoma, Norman, OK, USA. [13]Earth and Environmental Sciences, Lawrence Berkeley National Laboratory, Berkeley, CA, USA. [14]These authors contributed equally: Jing Tian, Jennifer A. J. Dungait, Ruixing Hou, Ye Deng. ✉e-mail: tianj@cau.edu.cn; zhangfs@cau.edu.cn; Francesca.Cotrufo@colostate.edu; jzhou@ou.edu

issue, there exists significant potential for SOC accrual in croplands through the adoption of 'climate-smart' management practices[5], which is considered a key natural solution for mitigating climate change and ensuring food security, and thereby achieving Sustainable Development Goals[6]. However, it is essential to recognize that SOC is vulnerable to loss due to climate change. Both soil warming experiments and global datasets have provided evidence of increased rates of decomposition under warming[7,8]. Unfortunately, studies investigating the interactive effects of management and warming on SOC accrual in croplands are extremely scarce. Understanding the interactions between management and warming is critical to identify suitable management practices that retain and augment SOC in face of a changing climate, and to develop effective strategies that increase agricultural resilience as a vital component of 'climate-smart agriculture'.

Soil disturbance caused by tillage has been a primary driver of historical SOC loss, estimated at 0.3–1.0 Pg C year$^{-1}$ globally[9]. In general, SOC accumulates after a shift from intensive tillage to conservation agriculture[10,11]. Conservation agriculture, typically represented by crop residue retention, and no-tillage or reduced tillage[12], has been proposed as an appropriate option for rebuilding SOC levels, which also provides various benefits for ecosystem multifunctionality[13,14]. It is estimated that 9–15% of global arable land has been managed by conservation agriculture approaches[12]. However, the response of SOC to alterations in tillage practices can differ significantly across different regions and over time[11,15]. Since increasing temperatures are expected to stimulate microbial respiration, soils with higher organic carbon contents are more vulnerable to carbon loss under warming conditions[8,16]. Soils managed under conservation agriculture, which involve increased retention of organic residue and reduced or zero tillage[14], should contain more SOC. They may also have greater proportions of relatively loosely protected chemically recalcitrant organic matter pools (e.g., macroaggregates or particulate organic matter), which are more vulnerable to loss under warming[7,17]. However, a short-term 3-month laboratory incubation study detected no significant difference in SOC mineralization between conservation and conventional agriculture under various temperature conditions[18]. The information regarding the sensitivity and persistence of SOC rebuilt through conservation agriculture management to long-term climate warming is very limited.

Predicting changes in SOC is highly dependent on microbial acclimation, which involves physiological adjustments that modify microbial carbon use efficiency (CUE) i.e., the proportion of carbon allocated to growth relative to respiratory losses[19–24]. Microbial CUE has strong implications for SOC storage in soils, as organic carbon converted into microbial biomass and necromass plays a crucial role in long-term SOC stabilization[25,26]. Microbial CUE depends on both abiotic and biotic factors[23], which are affected by warming and management practices. Increased temperature can have both direct and indirect effects on microbial CUE. Generally, warming decreases microbial CUE, as a greater proportion of the substrate is reallocated from growth to maintenance metabolism[19,27], which alters rates of enzyme-driven processes[28–30]. Warming can alter CUE indirectly via changes in soil moisture, substrate availability or the composition and/or structure of microbial communities[27,31]. Warming may decrease soil moisture and reduce microbial CUE because more substrate is allocated to dissimilatory metabolism, and hence less available for growth[19,32]. In contrast, warming may enhance plant growth, productivity and rhizosphere carbon input[33,34], promoting microbial growth and necromass accumulation[35]. The dominant microbial groups altered by warming could further modify community CUE and consequently SOC[36,37]. Microbial decomposers with higher CUE can convert substrates to new biomass more efficiently by increasing growth and reducing respiration per unit carbon taken up by microorganisms[38]. Thus, increasing CUE yields more microbial

biomass and byproducts, i.e., 'necromass', both of which contribute to the persistent SOC pool[39]. Fungi generally have a higher CUE than bacteria[40,41], but they can be negatively affected by tillage[42]. The greater sensitivity of fungi to warming compared to bacteria[37,43] may increase the contributions of fungal residues to necromass[44]. However, CUE may decrease if fungi allocate metabolic resources to increase exoenzyme production for nutrient acquisition under warming or nutrient limitation[45,46]. Crop residue retention in conservation agriculture can increase microbial CUE[47] and necromass accumulation[42,48] by alleviating nutrient limitation, preserving soil moisture, and reducing energy leakage pathways and exoenzyme production[49]. However, our understanding of the intricate connections between microbial physiological and metabolic characteristics and SOC accrual in response to management and warming is primitive, particularly when considering different temporal scales[50,51].

Herein, we present the study from a long-term agricultural experiment that spans a decade and encompasses two distinct management systems (conservation versus conventional agriculture) × two warming levels (warming versus ambient). Our primary objectives were to elucidate the interactive effects of warming and management on SOC accrual and persistence, as well as identify the temporal microbial attributes underpinning their responses over 10 years. Specifically, our study aimed to: (i) assess whether warming differentially affects SOC accrual under conservation agriculture (chopped crop residues returned and no tillage) versus conventional agriculture (crop residue removed and annual tillage); and (ii) examine the interactive effects of warming and management on the succession of microbial communities and temporal changes of microbial physiological traits (e.g., microbial growth, CUE, microbial necromass carbon, and microbial metabolic functional genes) and evaluate their consequences for SOC formation and accrual. We hypothesized that, under conditions of climate warming, (i) conservation agriculture increases SOC directly through increased plant-derived carbon inputs and indirectly via greater substrate availability to the soil microbial community; (ii) microbial community-level adaptation to warming and higher microbial growth efficiency and metabolic functions in response to larger substrate availability increases the contribution of microbial necromass to SOC over time under conservation agriculture; (iii) the increase in size and accelerated fungal community turnover boost fungal necromass accumulation, thereby promoting SOC formation and persistence over time under conservation agriculture. To test these hypotheses, we measured SOC, bacterial and fungal communities using DNA sequencing, microbial functions using metagenomics, and microbial physiological traits (CUE and microbial necromass) using substrate independent $H_2^{18}O$ labeling and microbial biomarker (amino sugars) analysis. In our study, long-term warming increased SOC under conservation agriculture, but not under conventional agriculture. This response under conservation agriculture was related to increase in key microbial physiological traits, CUE, growth and fungal necromass, with accelerated fungal community turnover and divergence over the 10 years. Using structural equation models analysis, we find that an increase in carbon input from the crops accelerated fungal succession and enhanced microbial growth efficiencies, leading to a progressive increase of microbially-derived carbon contributions to SOC formation and accrual at decadal time-scales under conservation agriculture with warming. Our work demonstrates that agricultural management and climatic factors can interact to alter microbial community composition, physiology and functions and, in turn, SOC formation and accrual in croplands.

## Results

### Conservation agriculture increased SOC by mediating the effect of warming on soil and plant properties

Warming the field plots under conservation or conventional agriculture was experimentally imposed for 10 years using infrared

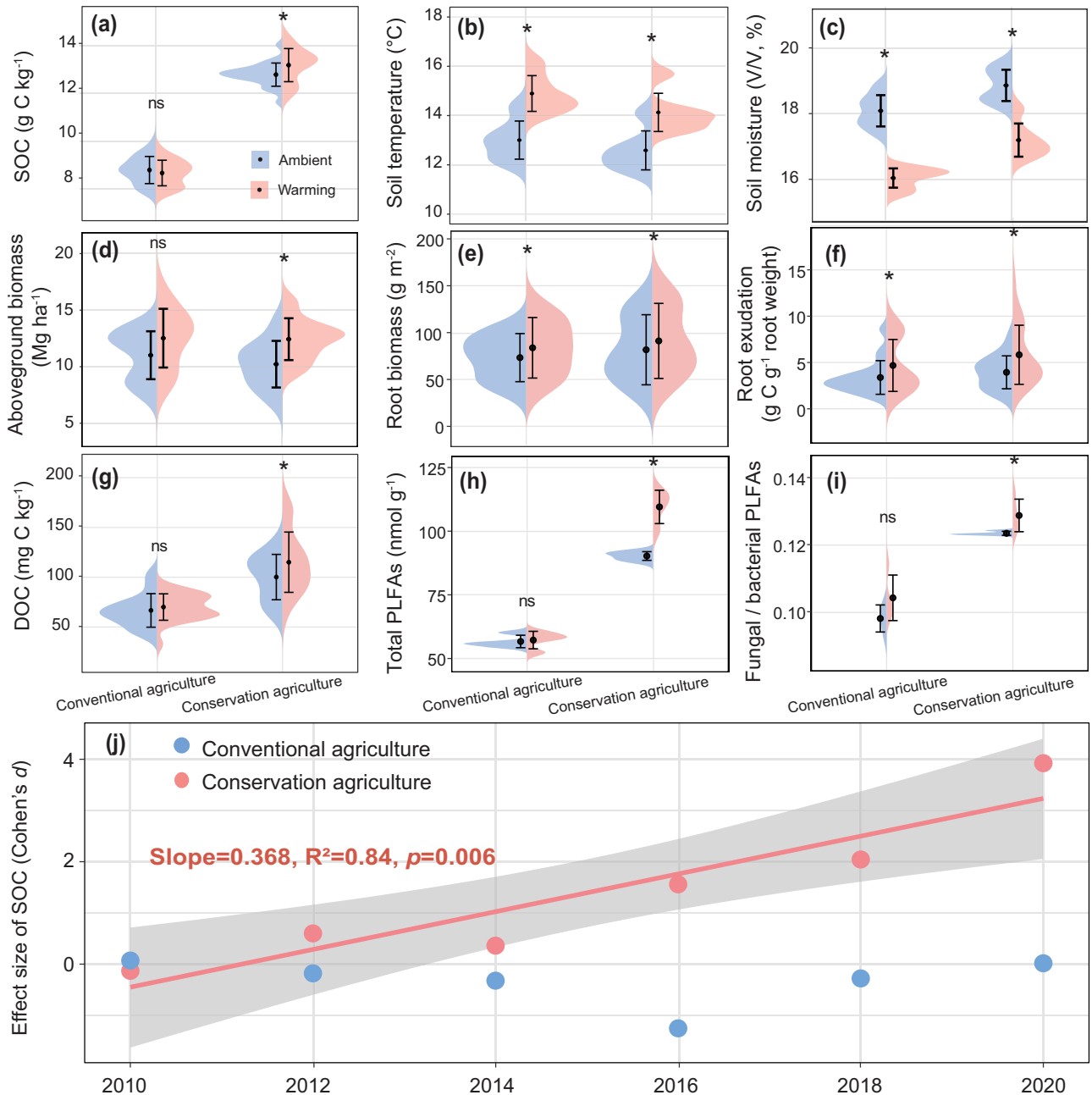

**Fig. 1 | Effect of warming on plant and soil variables over ten years depending on management systems. a–i** Average SOC content, soil temperature and moisture, aboveground biomass, root biomass, root exudation C, DOC and microbial biomass. Data are presented as violin plots with mean values ± s.e.m. The root biomass and root exudation C input and microbial biomass data were analyzed based on soil sampling in 2020 ($n = 4$ independent soil samples per treatment). The other parameters were average data on six sampling dates over 10 years ($n = 24$ independent soil samples per treatment). Statistical analysis was performed using repeated measures ANOVA analysis. All reported $p$ values result from two-sided statistical tests with *$p < 0.05$; ns nonsignificant. Asterisks indicate significant differences in the warming effect of the individual management system as compared

with their matched no-warming condition. For exact statistical values, see Supplementary Tables 2. **j** Shift in the effect size of warming on SOC over time (2010–2020) for conservation and conventional agriculture. Linear regression model with two-sided test was used for the statistical analysis, and adjusted R-squared was used. Relationships are denoted with solid lines and fit statistics (slopes, $R^2$ and $p$ values) for each management practice. The solid line represents the significant linear regression ($p < 0.05$), and the gray shading indicates the 95% confidence intervals. Conserv-Amb conservation agriculture without warming, Conserv-Warm conservation agriculture with warming, Conven-Amb conventional agriculture without warming, Conven-Warm conventional agriculture with warming.

heaters, maintaining the soil temperature at +2 °C above ambient levels (Supplementary Fig. 1). We first determined the direction of temporal changes in SOC and investigated whether warming differentially affected SOC content under conservation versus conventional agriculture. Over the course of 10 years, conservation management increased SOC content compared to conventional management, regardless of warming treatment. Warming further increased SOC

content under conservation agriculture by 3.1% as compared to the no-warming treatment, but not under conventional agriculture ($p < 0.05$; Fig. 1a; Supplementary Tables 1 and 2). Along with individual effects of management and warming, there were interactive effects of management × warming and management × warming × year on SOC content ($p < 0.05$; Supplementary Table 1), suggesting that the positive stimulatory effect of warming on SOC content was mediated by

management and such effect increased with time. We further assessed the warming effect over 10 years on SOC content using Cohen's *d* index: SOC content increased linearly with the duration of warming under conservation agriculture, and this increase accelerated after the 5th year (*p* = 0.006, 2016–2020; Fig. 1j). In contrast, the warming effect on SOC content under conventional agriculture was not significant (Fig. 1j; Supplementary Table 2). Therefore, our findings partially support the hypothesis (i), suggesting that conservation agriculture under warmer conditions increases SOC content.

The conservation agriculture treatment, characterized by continuous soil cover by crop residues with no tillage, increased SOC, which mitigated the effects of experimental warming on soil temperature and moisture contents. Over the 10-year study period, we continuously monitored soil temperature and moisture using in-field sensors. We observed that both of these factors were changed by management and experimental warming. As expected, experimental warming increased soil temperature but decreased soil moisture under both conservation and conventional agriculture (*p* < 0.05; Fig. 1b, c; Supplementary Tables 1 and 2). However, the warming effects were modulated by management. Soils covered by crop residues under conservation agriculture were cooler (1.5 °C vs. 1.9 °C; *p* < 0.05; Fig. 1b) and wetter (11% vs. 8.9 %, *p* < 0.05; Fig. 1c) compared with conventional agriculture.

We also evaluated plant carbon input using gravimetric measurements of aboveground biomass and root biomass, and root exudation carbon using a method developed for in situ collection of roots exudates[52,53]. Warming increased aboveground biomass under conservation agriculture, but not under conventional agriculture (*p* < 0.05; Fig. 1d; Supplementary Table 2), probably due in part to the positive effects of residue retention and no tillage on soil moisture. Warming increased belowground plant carbon inputs, including root biomass and root exudate carbon, in both conservation and conventional agriculture (*p* < 0.05; Fig. 1d, e, f; Supplementary Table 2). Overall, total root carbon input increased by 65% under warming compared with ambient control under conservation agriculture (*p* < 0.05; Supplementary Fig. 2), ultimately contributing to the increase in SOC (*p* < 0.05; Fig. 1a; Supplementary Table 2) and dissolved organic carbon (DOC) (*p* < 0.05; Fig. 1g; Supplementary Table 2) concentrations.

### Warming stimulated microbial growth efficiency and fungal necromass carbon accumulation under conservation agriculture

Like SOC content, the effects of management and warming on microbial growth efficiency might change over 10 years. To address this, we measured soil microbial community CUE using the substrate-independent ¹⁸O-H₂O method[54,55]. Across all years of the study, warming increased microbial CUE by 12%, microbial growth by 43% and carbon uptake by 24% when compared with the ambient control under conservation agriculture (*p* < 0.05; Fig. 2a; Supplementary Tables 1 and 2), indicating an acceleration of microbial turnover including proliferation, growth and death. The interactions between management × warming and management × warming × year on microbial CUE, growth, respiration and carbon uptake (*p* < 0.05; Supplementary Table 1) indicated that warming effects were dependent on both management and time. We further examined the temporal changes of the warming effect on microbial growth efficiency under the two management systems over the 10-year study period (Fig. 2b). The warming effects on microbial CUE under conservation agriculture increased linearly with experimental duration (Slope = 0.83, *p* = 0.01; Fig. 2b).

To better understand these effects, we divided the study into early (2010–2015) and later stages (2016–2020) to assess the role of warming duration on the response directions and magnitudes (Supplementary Fig. 3). Shorter and longer-term warming had contrasting

effects on CUE under conservation agriculture: a negative effect size of CUE was observed during the early stage (2010–2015), while a positive effect size was evident in the later stage (2016–2020) (*p* < 0.05; Supplementary Fig. 3a). Consequently, warming increased microbial CUE by 1.1–1.5 times particularly under conservation agriculture during the later stage (*p* < 0.05; Supplementary Fig. 3e). The warming effect on microbial growth and carbon uptake followed a similar pattern to that of CUE, with the difference between warming and no-warming increasing over time in soil under conservation agriculture (Slope = 0.58 and 0.44, all *p* < 0.05; Fig. 2b; Supplementary Fig. 3b, d). Conversely, no effects were observed under conventional agriculture (Fig. 2b; Supplementary Fig. 3). The observation that microbial CUE, growth and carbon uptake were enhanced in the later stage of the experiment indicated greater substrate availability, which agrees with hypothesis (i) as plant carbon inputs and SOC content increased more rapidly after the 5th year (Fig. 1j).

The acceleration of microbial turnover in response to warming under conservation agriculture was substantiated by a 77% increase in total microbial necromass carbon (indicated by the concentration of amino sugar biomarkers[56]) compared with un-warmed soils (*p* < 0.05; Fig. 2c; Supplementary Tables 2 and 3). Warming elevated the contribution of total necromass to SOC during the later stage of the experiment (from 2016 to 2020; *p* < 0.05; Supplementary Fig. 4), underscoring the important role of the entombing effect of the microbial carbon pump in the formation of new SOC. In particular, fungal necromass carbon increased linearly with year under conservation agriculture, with the effect size of warming being 19 times greater in 2020 than in 2010 (Slope = 1.6, *p* = 0.04; Fig. 2d; Supplementary Table 4). However, this effect was not significant under conventional agriculture. A similar temporal pattern was observed for total microbial necromass carbon (R² = 0.81, *p* < 0.03; Fig. 2d). An increase in the concentration of microbial PLFA biomarkers, which serve as indicators of microbial biomass of dominant groups (Fig. 1h; Supplementary Table 2), also indicated the benefits of conservation management and warming for fungi, as reflected in larger fungal/bacterial PLFA ratios (Fig. 1i; Supplementary Table 2). The larger fungal biomass consequently led to larger fungal necromass, contributing significantly to total necromass over time (*p* < 0.05; Supplementary Fig. 4) and was further increased by warming (by 29%; *p* < 0.05; Fig. 2c). Correspondingly, warming increased the contribution of total and fungal necromass to SOC from 33% to 61%, and 28% to 53%, respectively, from 2010 to 2020 under conservation agriculture (*p* < 0.05, Supplementary Fig. 4). The increase in microbial growth efficiency and total necromass under conservation agriculture in warmed soils, in parallel with larger plant carbon input and increasing SOC content, suggests that the microbial community and its physiological responses adapted to warming conditions over time, which supports the hypothesis (ii).

### Warming accelerated fungal community temporal turnover and its divergence under conservation agriculture

To understand the role of the soil microbiome in driving SOC dynamics in response to warming and management, we examined how microbial diversity and community turnover co-varied over the 10 years using DNA sequencing. Warming altered both microbial communities and individual microorganisms, depending on management. The effects of warming on fungal phylotypes changed with year under conservation agriculture (*p* < 0.05; Supplementary Fig. 5). In the later stage of the experiment from 2016 to 2020, warming decreased the fungal phylotypes by 9.3–12% under conservation agriculture, but had no effect under conventional agriculture (*p* < 0.05; Supplementary Fig. 5). The compositions of soil bacteria and fungi were altered by warming, management and year, as visualized by principal component analysis (Supplementary Fig. 6; Supplementary Tables 5). Three complementary non-parametric multivariate statistical tests (Adonis,

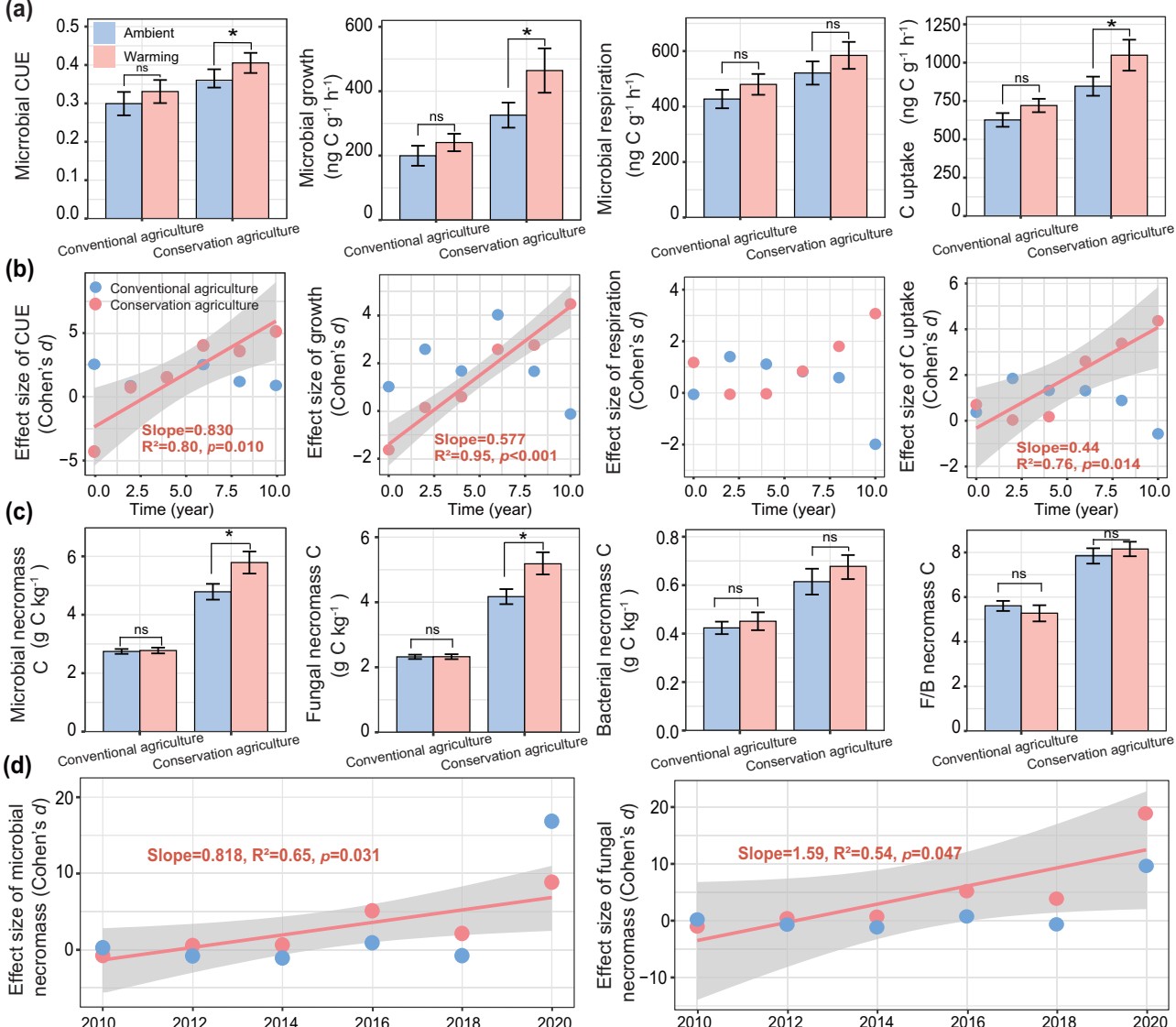

**Fig. 2 | Effects of warming on soil microbial CUE, growth, respiration, C uptake and necromass C over ten years depending on management systems. a** Average microbial CUE, growth, respiration and C uptake on six sampling dates over 10 years. **b** Shifts in the effect size of warming on microbial CUE, growth, respiration and C uptake over time (2010–2020) for conservation and conventional agriculture. **c, d** Average microbial necromass C and shift in the effects of warming on necromass over time (2010–2020) depending on the management systems. Letters indicate significant differences between warmed and control under two management systems. Bars represent mean ± s.e.m. ($n = 24$ independent soil samples per treatment). Statistical analysis was performed using repeated measures ANOVA

analysis. The two-sided statistical tests indicate significant effects by *$p < 0.05$; ns, nonsignificant. For exact statistical values, see Supplementary Tables 2 and 4. Relationships are denoted with solid lines and fit statistics (slopes, $R^2$ and $p$ values) for each management practice (adjusted r-squared and $p$ value are shown). The solid line represents the significant linear regression ($p < 0.05$), and the gray shading indicates the 95% confidence intervals. Conserv-Amb conservation agriculture without warming, Conserv-Warm conservation agriculture with warming, Conven-Amb conventional agriculture without warming, Conven-Warm conventional agriculture with warming.

Anosim and MRPP) confirmed that bacterial and fungal communities across all years were significantly different between the warmed and unwarmed plots under both conservation and conventional agriculture ($p < 0.05$; Supplementary Table 6). Under conservation agriculture, warming had a large negative effect on the relative abundance of Acidobacteria ($\beta = -0.122$, $p < 0.001$; Fig. 3a), while it increased the relative abundances of Actinobacteria, Firmicutes, Bacteroidota, and Verrucomicrobiota ($\beta = 0.002$ to 0.07, $p < 0.05$; Fig. 3a). The relative abundance of Proteobacteria, including Alphaproteobacteria, increased under conventional agriculture with warming ($p < 0.01$; Fig. 3a). Warming had a greater positive effect on the relative abundance of Ascomycota under conservation than conventional agriculture ($\beta = 0.026$ vs 0.009, $p < 0.05$; Fig. 3a). Warming effects on the

relative abundance of Ascomycota shifted from negative in the early stage ($\beta = -0.034$; Supplementary Fig. 7; from 2010 to 2015) to positive in the later stage ($\beta = 0.085$; from 2016 to 2020). We further investigated different fungal guilds as classified by FUNGuild. While warming marginally increased the relative abundance of Saprotrophs under both management types, the effect was more pronounced under conservation agriculture ($\beta = 0.015$ vs 0.055; Fig. 3a).

To elucidate the impacts of warming on the temporal turnover of microbial community structure, we assessed the time-decay relationships (TDRs) for bacteria and fungi. The slopes of the TDR values represent the temporal turnover rates of soil microbial communities[57]. Temporal turnover rates of fungi were faster than those of bacteria ($v = 0.24$–0.35 VS 0.72–0.97; $p < 0.001$; Fig. 3b). Moreover, the TDR

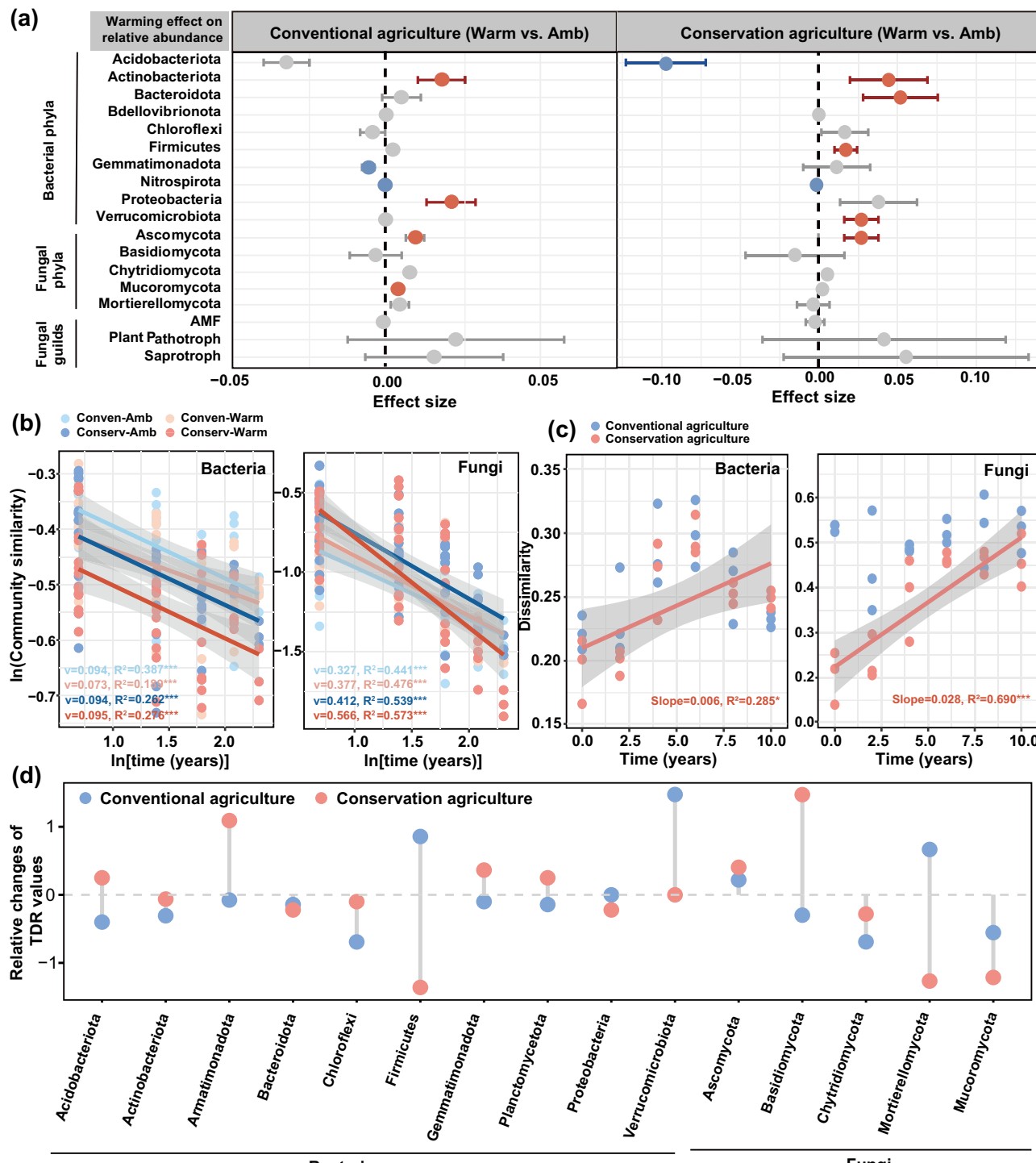

**Fig. 3 | Temporal dynamics of the soil microbiome. a** Effect sizes of warming on the relative abundance of major microbial groups based on linear mixed-effects models. Data are presented as mean ± s.e.m. of the estimated effect sizes. Statistical significance is based on Wald type II χ2 tests (two-sided) (n = 18 paired soil samples per management); non-significant changes are denoted by gray dots. **b T**he time-decay relationships of bacterial and fungal communities for all treatments and (**c**) temporal changes in microbial community differences between warmed and control soils under conservation and conventional agriculture. Linear regression model with two-sided test was used for the statistical analysis, and adjusted R-squared was used. Relationships are denoted with solid lines and fit statistics (slopes, $R^2$ and $p$ values) for each management practice. The solid line represents the significant linear regression ($p < 0.05$), and the gray shading indicates the 95% confidence intervals. **d** The warming-induced relative changes of TDRs (v) among phylogenetic lineages under conservation and conventional agriculture. The TDR values are presented as the relative changes of (warming-control)/control. Conserv-Amb: conservation agriculture without warming; Conserv-Warm: conservation agriculture with warming; Conven-Amb: conventional agriculture without warming; Conven-Warm: conventional agriculture with warming.

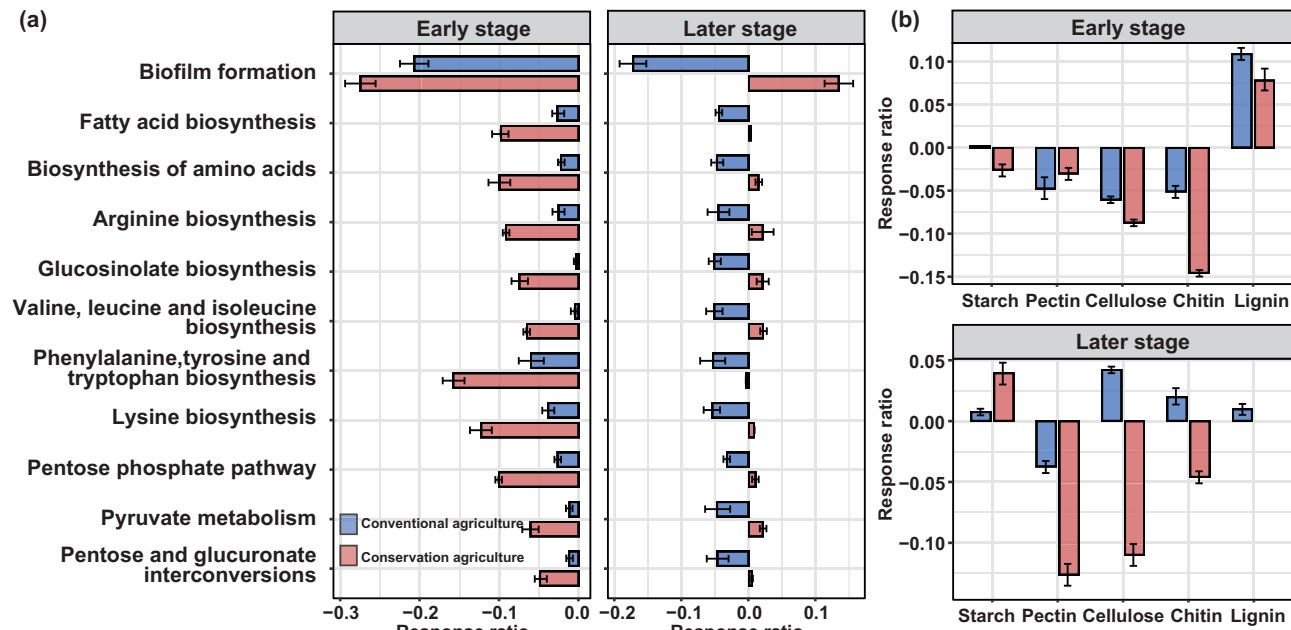

**Fig. 4 | Effect of warming on carbon metabolism-associated functions of soil microorganisms depending on management at early and later stages. a** Effect of warming on microbial growth-related pathways under two management systems based on the KEGG database. Only significant changed pathways between warmed and control soils under each management were presented here. **b** Effect of warming on the relative abundance of carbohydrate metabolism genes under two management systems based on carbohydrate-active enzymes (CAZy). Only significant changed genes were presented here. Data are presented as mean ± s.e.m. of the estimated effect sizes ($n = 3$ paired soil samples per management). Conserv-Amb: conservation agriculture without warming; Conserv-Warm: conservation agriculture with warming; Conven-Amb: conventional agriculture without warming; Conven-Warm: conventional agriculture with warming.

slopes were steeper in response to warming under conservation agriculture (pairwise permutation tests; $p < 0.001$; Fig. 3b, Supplementary Table 4). To further examine the warming effects on the succession of bacterial and fungal communities over time, we assessed the differences in microbial communities between paired warmed and unwarmed plots under these two managements on a yearly basis. The differences in bacterial and fungal community structures between warming and ambient control increased linearly with time, but only under conservation agriculture ($p < 0.05$; Fig. 3c). In addition, the paired differences in microbial communities between the warmed and control plots under conservation agriculture were more pronounced for fungi (slope = 0.006; Fig. 3c) than for bacteria (slope = 0.028; Fig. 3b), suggesting that warming had a more substantial impact on the temporal turnovers of fungi than bacteria under conservation agriculture, agreeing with the hypothesis (iii) that warmer soils under conservation agriculture accelerated fungal community turnover.

We also observed that fungal groups and community turnover changed in response to warming under conservation agriculture, which were in parallel with changes in microbial growth efficiency and necromass accumulation (Fig. 2). To further explore these dynamics, we evaluated the TDRs of various lineages of bacteria and fungi between warmed and unwarmed plots under both management types (Fig. 3c). The TDRs of fungal phyla were higher than bacterial phyla ($v = 0.02$–$0.23$ vs. $0.04$–$0.89$; Supplementary Table 7). The TDR slopes for Basidiomycota and Ascomycota communities differed between warming and no-warming under conservation agriculture ($p < 0.001$; Supplementary Tables 7 and 8). Warming accelerated the temporal turnover rates of Ascomycota and Basidiomycota under conservation agriculture, but decreased Firmicutes and Planctomycetota communities ($p < 0.05$; Fig. 3c; Supplementary Table 7). The threshold indicator taxa analysis (TITAN) also showed that Ascomycota and Basidiomycota had large increases over time (Supplementary Fig. 9). Moreover, warming had more pronounced effects on the TDRs of bacterial taxa, including

Firmicutes and Gemmatimonadota under conventional agriculture ($p < 0.05$; Fig. 3c; Supplementary Table 7).

## Warming effects on microbial functions related to microbial growth depended on management

Management and warming changed the magnitudes of plant carbon inputs, including crop residues from aboveground biomass, root biomass and root exudates (Fig. 1). These changes had cascading effects on the biomass, activity (Figs. 1h, i, 2), structure and turnover (Fig. 3) of the soil microbial communities, particularly the proliferation of fungi. We further used shotgun metagenomic analysis to determine the changes in microbial functions related to cellular processes and metabolism (Fig. 4a). The relative abundances of pathways involved in the biosynthesis of microbial cell constituents were decreased in response to warming under both conservation and conventional agriculture at the early stage (in 2010) ($p < 0.05$; Fig. 4a). However, these pathways increased under conservation agriculture with warming at the later stage (in 2020), especially for biofilm formation and amino acids biosynthesis, suggesting a potential stimulation of gene activity associated with protein production and cell growth potential ($p < 0.05$; Fig. 4a). To delve deeper into critical processes associated with microbial membrane structure, we further analyzed the response ratio of KOs related to biofilm formation, fatty acid biosynthesis and biosynthesis of amino acids to warming under both management types ($p < 0.05$; Fig. 4b). Similar to the variations in metabolic pathways, the relative abundances of functional genes annotated in biofilm formation showed the greatest positive response to warming under conservation agriculture at the later stage, along with genes associated with fatty acid biosynthesis and biosynthesis of amino acids ($p < 0.05$; Supplementary Fig. 8).

Changes in the genes encoding carbohydrate-active enzymes based on the CAZy database (Fig. 4b) provided insights into how management and warming affected plant carbon inputs, including lignin, cellulose and pectin from plant cell walls, and microbial inputs such as chitin from fungal biomass (indicated by the increase in fungal glucosamine; Fig. 2c). Warming increased the relative abundances of

genes encoding ligninolytic enzymes by 7.8–11%, but decreased those of genes encoding pectin, cellulose and chitin under both conservation and conventional agriculture at the early stage ($p < 0.05$; Fig. 4b). Some distinct responses to warming were found between the management practices at the later stage. Under conservation agriculture, warming increased the relative abundances of genes encoding starch-degrading enzymes by 4.0%, yet decreased the relative abundances of genes associated with pectin, cellulose, and chitin degradation ($p < 0.05$; Fig. 4b). This suggested that conservation agriculture tends to favor more the retention of recalcitrant carbon (e.g., lignin and chitin) than labile carbon (e.g., starch) under warming, in agreement with the hypothesis (ii). In contrast, warming promoted the relative abundances of genes associated with starch, cellulose, chitin and lignin degradation by 0.76%, 11%, 4.60%, and 6.77%, respectively, under conventional agriculture in the later stage ($p < 0.05$; Fig. 4b).

### SOC accumulation as a function of warming-induced substrate availability, microbial community and physiological traits

Structural equation modeling (SEM) was used to determine the warming-induced abiotic (microbial physiology and diversity) and biotic (substrate availability and microclimate factors) components that drive SOC dynamics. Our results revealed the crucial role of soil microbial physiological traits in mediating SOC dynamics under conservation agriculture. A total of 85% of the SOC variations were accounted for by the combinations of soil temperature, substrate availability (indicated by plant biomass and DOC), microbial alpha diversity, microbial community composition, microbial CUE and necromass carbon (Fig. 5a). Among these factors, the positive direct effect of fungal necromass carbon on SOC was the greatest (Fig. 5b), while microbial CUE were mainly related to indirect effects (Fig. 5b). This corresponded to the strong positive correlation between fungal necromass carbon and SOC ($p < 0.05$; Supplementary Fig. 9a). The direct positive effect of microbial CUE on fungal necromass carbon was the largest (path coefficient = 0.64; Fig. 5b): fungal necromass carbon increased at a faster rate with microbial growth than respiration, leading to a significant positive relationship between fungal necromass carbon and CUE ($p < 0.05$; Supplementary Fig. 9b). Substrate availability exerted a strong selection on the fungal community composition through direct effect. Overall, fungal community composition showed positive associations with microbial growth ($p < 0.05$; Fig. 5c; Supplementary Table 9), leading to strong positive direct effects on microbial CUE (path coefficients = 0.47, Fig. 5a). The relative abundance of dominant fungal phylum Ascomycota showed negative correlations with microbial CUE and growth during early stage, but positive correlations in the later stage ($p < 0.05$; Supplementary Fig. 9d). The relative abundance of fungal phylum Basidiomycota had positive associations with CUE and microbial growth ($p < 0.05$; Fig. 5c; Supplementary Table 9). Moreover, microbial CUE and necromass increased with the relative abundances of genes related to microbial growth potential but decreased with genes encoding carbohydrate-active enzyme degradation ($p < 0.05$; Fig. 5d; Supplementary Table 10).

## Discussion

This study investigates the interactive effects of management and warming on SOC accrual and persistence in arable land over a decade-long timescale. Our findings provide empirical evidence that conservation agriculture can increase the accrual and persistence of SOC under predicted climate warming scenarios. SOC dynamics is controlled by the balance between carbon input from plants and decomposition via microbial respiration, both of which are modulated by management and climatic factors[58]. Generally, warming per se is anticipated to cause SOC loss, particularly in soils with high organic carbon contents[7,8]. Soils managed under long-term conservation agriculture often harbor substantial pools of unprotected organic carbon (e.g., particulate organic matter), which may be more

vulnerable to warming[17]. We did not observe that warming affected the SOC content under conventional agriculture (Fig. 1a), but the relative abundances of microbial genes encoding labile and recalcitrant carbon degradation were increased by warming, suggesting the potential for further SOC loss (Fig. 4b). Though warming increased belowground biomass and root exudation (Fig. 1e–f), these changes were not translated to an increase in SOC under warmed conventional agriculture. Herein, SOC content increased exclusively under conservation agriculture and was further boosted by 10 years of warming (Fig. 1a, j), which can be partially attributed to continuous soil cover that mediated the effects of warming on soil temperature and prevented excessive drying. Importantly, microbial physiological traits and community turnover shifted after the 5th year of warming under conservation agriculture (Figs. 2 and 3; Supplementary Figs. 3 and 4), leading to higher SOC accumulation (Fig. 1j). These findings suggested that more efficient soil microbial carbon cycling underlies the observed SOC accumulation under conservation agriculture in response to warming, especially after 5 years.

Microbial CUE is a crucial metabolic parameter that describes the ratio of anabolic and catabolic processes[19]. Higher microbial CUE in the conservation agriculture with warming treatment, characterized by a greater increase in microbial growth (33%) compared to respiration (13%) over 10 years (Fig. 2a, b and Supplementary Fig. 3), indicates an overall enhancement in microbial growth efficiency. The increase in microbial CUE under conservation agriculture incorporates the responses of maintenance costs and growth rates to warming-induced changes in soil temperature, moisture, substrate availability and microbial communities in this study. The fungal community structure exerted the strongest positive influence on microbial CUE, though the relationship was mainly mediated by substrate availability (Fig. 5a, b). Root biomass and root exudation are major forms of carbon input into soils, which serve as important carbon sources for soil microorganisms. Together with crop residue retention, DOC, aboveground biomass, belowground root biomass and rhizodeposition (Fig. 1d–g) increased under long-term conservation agriculture with warming, indicating higher plant carbon inputs (Fig.1d–f) that provided abundant substrates for microbial proliferation. This particularly favored fungal growth, leading to higher fungal biomass (Fig. 1h) and fungal/bacterial ratios (Fig. 1i) under long-term conservation management with warming. Fungi are sensitive to changes in local climate (e.g., moisture and temperature) and external disturbance[59]. Conservation agriculture promotes fungal abundance by increasing substrate availability, reducing disturbance of soil structure, preserving hyphal networks[60] and improving water availability[14]. Microbial CUE in response to warming increased linearly with time under conservation agriculture, especially during the later stage from 2016 to 2020 (Fig. 2b and Supplementary Fig. 3). This also corresponded to an increase in the relative abundances of microbial functional genes associated with microbial cell growth and protein production potential (e.g., biofilm formation, fatty acid and amino acid biosynthesis; Fig. 4a), with fewer genes related to recalcitrant carbon decomposition under conservation agriculture with warming during the later stage (Fig. 4). This agrees with a long-term warming study in a tall-grass prairie, which reported a shift towards the decomposition of more labile substrates due to increased plant inputs[61]. This is also consistent with the observation that microorganisms may downregulate their protein production machinery in response to warming-induced substrate depletion[30]. Conversely, the utilization efficiency of more recalcitrant substrates was increased at higher temperatures in soils exposed to almost two decades of warming at 5 °C above ambient at Harvard Forest[20]. Altogether, our results underscore the importance of microbial community structure and substrate availability in driving the responses of microbial metabolic processes to warming under conservation agriculture.

Studies of soil microbial community turnover and microbial physiological traits over time in response to climate change are vital to

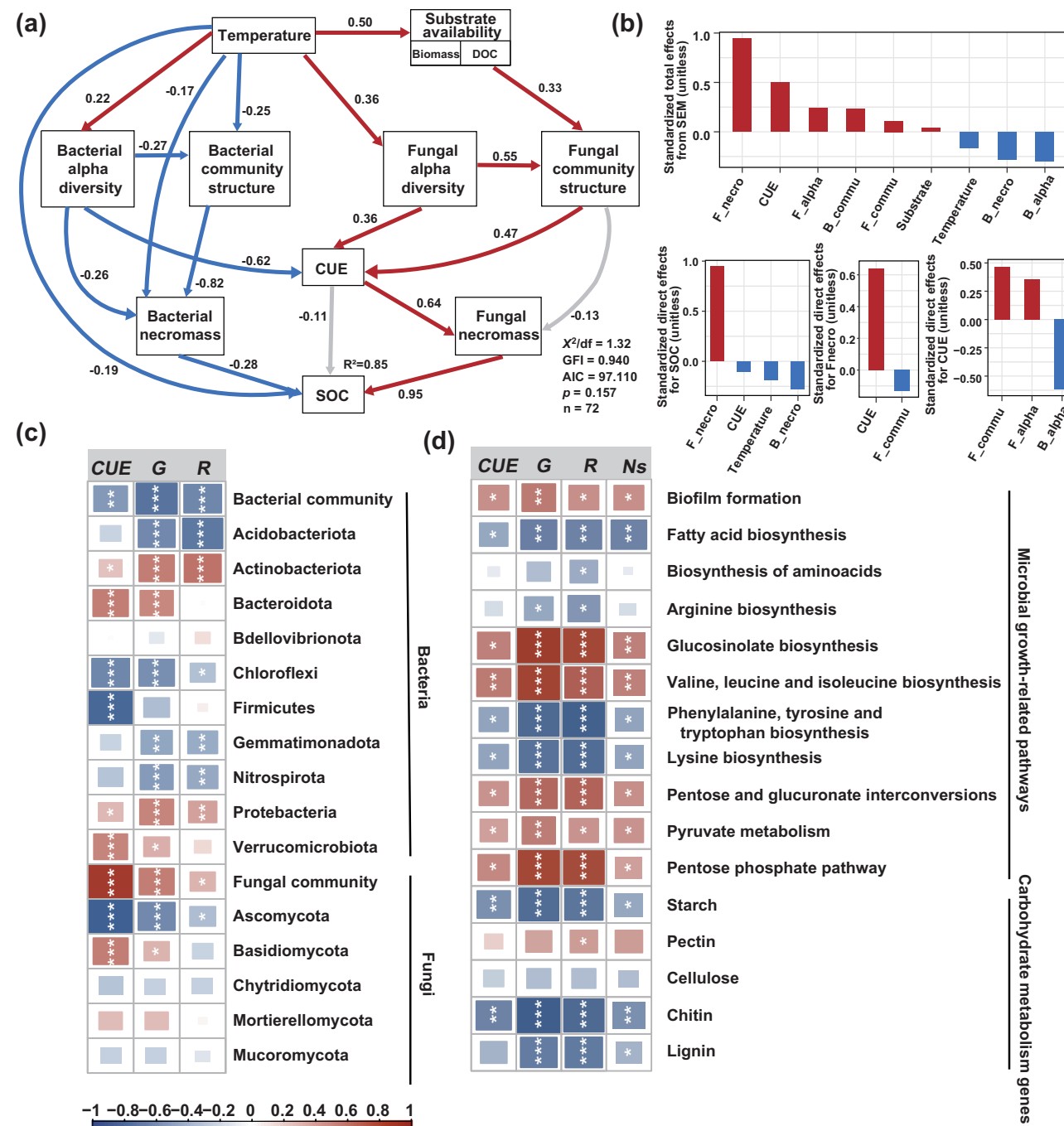

**Fig. 5 | The abiotic and biotic drivers of SOC content.** Structural equation model (SEM) for SOC content, showing the relationships between microclimate, substrate availability, microbial diversity, physiological traits and SOC (**a**). Significant paths are shown in red if positive or in blue if negative. Gray arrows indicate insignificant relationships. Path width corresponds to the degree of significance as shown in the lower left. Numbers near the pathway arrow indicate the standard path coefficients. The portion of variance explained by the model ($R^2$) is shown for response variable. **b** Standardized total and direct effects derived from the SEM depicted above. **c** Correlations between microbial community and its phylogenetic lineages richness and physiological traits. **d** Correlations between carbon metabolism-associated functions of soil microorganisms and physiological traits. The color denotes the correlation coefficient determined by the linear mixed-effects model. Statistical significance is based on Wald type II $\chi^2$ tests (two-sided) with $n = 72$ independent soil samples. Asterisks indicate significant differences (***$p < 0.001$, **$p < 0.01$, *$p < 0.05$). For exact $p$ values, see Supplementary Tables 9, 10. F-necro: fungal necromass C; B-necro: bacterial necromass C; F-alpha: fungal richness; B-alpha: bacterial richness; F-commu: fungal community composition; B-commu: bacterial community composition; G: microbial growth; R: microbial respiration; Ns: microbial necromass.

understand the long-term consequences of warming for SOC stocks, but are surprisingly sparse[62]. Herein, the soil fungal community exhibited the fastest temporal turnover, indicated by the steepest slope in the time decay plot, under conservation agriculture with warming (Fig. 3b). This finding challenges the assumption that soil bacterial community composition displays larger temporal turnover

than that of fungi due to the faster life cycles and lower metabolic cost of biosynthesis among bacteria[63]. Our results indicated that the turnover rates of fungal communities were approximately 2.1–3.5 times faster than those of bacterial communities (Fig. 3b), consistent with previous observations that the temporal succession rates of fungal communities increased more rapidly than those of bacteria in

response to warming in grassland[57] and a forest[64]. Fungi have a greater carbon demand than bacteria and are usually the first to utilize belowground plant carbon inputs[62]. Consequently, the rapid fungal growth and fungal community changes, driven by increased plant carbon input, are key drivers of temporal community turnovers under conservation agriculture with warming. Larger substrate availability due to increased plant carbon input had a direct effect on fungal community composition, as supported by the observed changes in fungal lineage (Fig. 5a–c). Among various fungal taxa, Ascomycota, previously reported to increase with warming[65], explained the largest variance in total fungal communities (Supplementary Fig. 6). As copiotrophic fungi, Ascomycota are generally *r*-strategists that efficiently utilize labile carbon sources for rapid metabolism, growth and propagation[66]. In response to increased substrate availability, these fungi may have a shorter generation time, leading to more generation per unit time[67]. Therefore, we assume that warming played a role in driving fungal temporal divergence, as evidenced by increasing differences between warmed and un-warmed plots under conservation agriculture (Fig. 3b). In contrast, Acidobacteria explained the largest variance for bacterial communities (Supplementary Fig. 6). This group is characterized by oligotrophic traits, meaning they thrive in low-nutrient environments[68] and are closely associated with decreases in net nitrogen mineralization due to the reduced substrate availability caused by soil drying[66]. Therefore, the slower temporal turnover of bacterial communities may be attributed to a large portion of inactive (dormant or slow-growing) bacteria in warmer and drier soils[69]. These results suggest that microbial physiological adjustments that selected microbial lineages were better adapted to changing soil microclimatic conditions and substrate availability over time.

The evolving theory of soil organic matter formation emphasizes microbial necromass production and mineral-matrix interactions as dominant mechanisms for SOC formation and stabilization[2,48]. Microbial necromass accounts for more than 50% of SOC, which is protected either physically within soil aggregates or chemically through its association with soil minerals[70]. Fungal necromass carbon increased with warming and accounted for the largest variations in SOC accumulation under conservation agriculture: 36% of total SOC was derived from microbial necromass (amino sugars), with ~86% of fungal origin (Figs. 3c, d and 5a). These findings support the concept that fungal necromass is a critical driver of stable SOC accrual[48,71]. Microbial necromass accumulation in soil is driven by plant carbon input and mediated by microbial necromass production and stabilization[35]. Accordingly, the fungal community was positively correlated with microbial CUE, which had the largest positive effects on fungal necromass (Fig. 5a, b). These results suggested that warming increased microbial growth efficiency and fungal community turnover due to greater plant carbon input, which ultimately increased the supply of microbial residues contributing to SOC formation. In addition, the adoption of conservation agriculture reduces soil disturbance and promotes aggregate formation and stability, providing a conducive environment for the preferential accumulation of fungal-derived necromass in macroaggregates[72]. Overall, our results suggested that warming enhances SOC formation and accumulation under conservation agriculture by increasing microbial growth efficiency via accelerating fungal community turnover, as well as via increased fungal necromass carbon accumulation (Figs. 2 and 3). In summary, this study provided empirical evidence for the mechanisms that enable the increase in SOC accrual and persistence under warming and conservation agriculture. As illustrated by Fig. 6, by carrying out a long-term in situ management × warming field experiment, we revealed a gradual acceleration of SOC accrual in response to climate warming under conservation agriculture. Our research highlights the significant roles of altered plant carbon input and microclimate conditions in driving microbial responses to warming under conservation

agriculture. In particular, under long-term conservation agriculture with warming, the increase in plant carbon input accelerated fungal succession and enhanced microbial growth efficiencies, leading to a progressive increase of microbially-derived carbon contributions to SOC formation and accrual at decadal timescales. If our findings can be generalized to other systems and regions where water does not limit productivity (e.g., irrigated regions), then we propose that regenerative management can promote effective carbon sequestration under warming, which increases agricultural resilience as a vital component of 'climate-smart agriculture'. However, more research is needed to understand the combined effects of management and climate on SOC accrual and persistence in arable soils across different regions with a wide precipitation range, which could guide climate-smart land use practices to sequester carbon effectively and optimize crop production in the face of a changing climate.

## Methods

### Experimental site and design

A long-term field experiment with a double-cropped winter wheat (*Triticum aestivum* L.)-summer maize (*Zea mays* L.) system was started in 2003 at the Yucheng Comprehensive Experiment Station in North China (36°51′N, 116°34′E), which belongs to the Chinese Academy of Science (CAS). The region has a temperate semi-arid climate with an annual mean temperature of 13.1 °C, and annual mean precipitation of 561 mm. The soil has a silt loam texture with 12% sand, 66% silt, 22% clay, and a mean pH of 7.1. The soil type is Calcaric Fluvisol according to the FAO-UNESCO system.

Four treatments were laid out in a randomized complete block design with four replicates: conservation agriculture with and without warming (Conserv-Amb, Conserv-Warm) and conventional agriculture with and without warming (Conven-Amb, Conven-Warm). Full details of the experimental design and management are provided by ref. 73. In brief, the conventional and conservation agriculture treatments were established in 2003. Two levels of warming (ambient and +2 °C) were imposed on both conservation and conventional agriculture since 2010. An infrared heater (MSR-2420 infrared heater, Kalglo Electronics Inc, Bethlehem, PA) was suspended approximately 3 m above the ground in each warmed plot to achieve a surface soil warming of 2 °C, which was projected by IPCC greenhouse gas scenarios rates for northern China.

The size of each plot was 2 m × 2 m. Winter wheat was seeded in early October and harvested in early June. The summer maize was grown in June and harvested at the end of September. In the conventional agriculture treatment, the residues were removed after the maize harvest. Cultivation with a rotary tiller to a depth of 10–15 cm fully incorporated the remaining stubble into the soil. In the conservation agriculture treatment, all residues were chopped to approximately 5 cm in length and retained on the soil surface. All other management procedures were the same for both management types.

### Field measurements, soil sampling and analyses

Soil temperature at 5 cm depth was monitored with PT 100 thermocouple. Volumetric soil moisture at 0–10 cm depth was measured by FDS100 soil moisture sensors (Unism Technologies Incorporated, Beijing). From 2010 to 2020, the living aboveground biomass was harvested (2 m × 2 m) at each plot in May every year. The aboveground plant biomass samples were dried at 80 °C until a constant weight was obtained. Measurement of root biomass and root exudation C was conducted from October 2019 to May 2020 during winter wheat growth seasons according to ref. 52. (details below).

Soil samples (0–5 cm depth) were collected every 2 years after winter wheat harvest from 2010 to 2020. Five topsoil cores taken randomly from each plot were composited as one sample per plot. The composited soil samples were then passed through a 2 mm sieve to remove visible roots and gravel. The sample was divided into 3 sub-

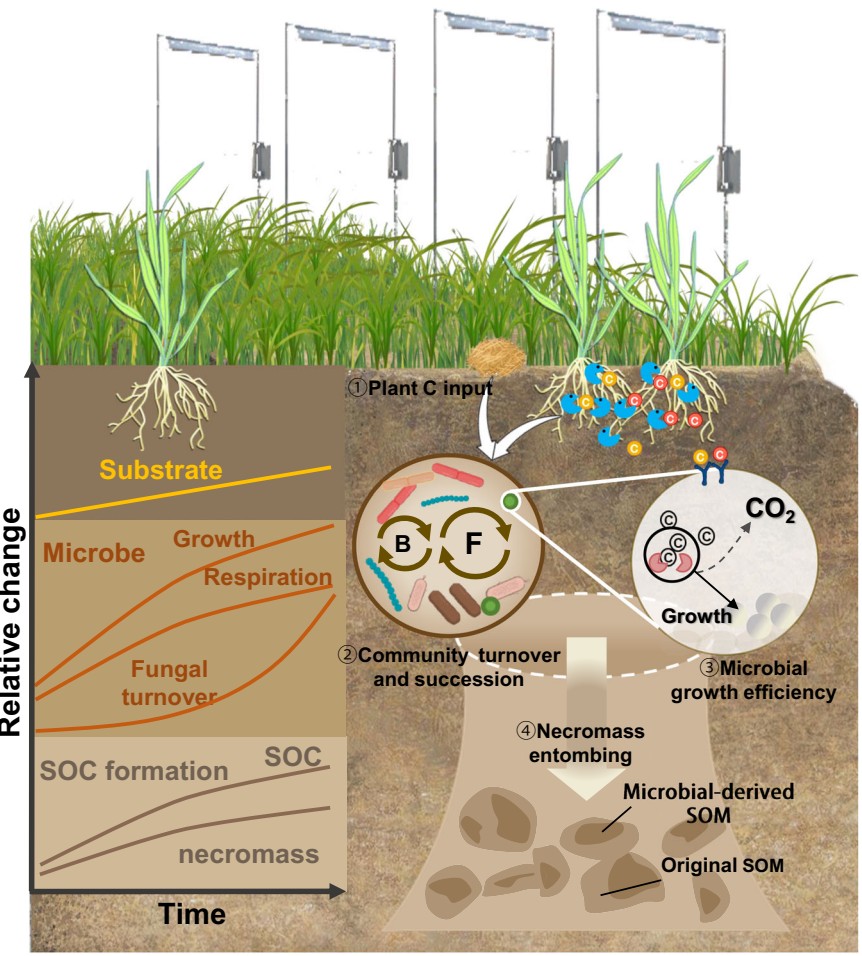

**Fig. 6 | A conceptual diagram illustrating the impact of warming on SOC formation and accrual and the underlying microbial mechanisms through plant C inputs, community succession, physiology adjustment and necromass production under conservation agriculture over decadal timescales.** The main processes include (1) substrate availability increased due to plant C input (e.g., crop residues from aboveground biomass, root biomass and root exudates) with warming; (2–3) the above processes worked as 'trigger' for driving microbial community succession (observed by high temporal scaling rates and increasingly divergent succession for fungal community) and microbial physiological adjustments (e.g., high CUE and growth efficiency). In parallel, the altered microbial metabolism-associated functional genes were observed indicated by an increased relative abundance of genes involved in microbial growth potentials and labile C degradation, but decreased genes encoding recalcitrant C degradation; (4) gradually strengthened microbial carbon pump observed as microbial necromass (especially fungal necromass) enters the carbon pools and drive new carbon formation and accrual.

samples for analyses. The first sub-sample was air-dried at room temperature, the second sub-sample was stored at 4 °C, and the third sub-sample was stored in a freezer at −80 °C. SOC, total nitrogen (TN), and DOC were measured after sample collection in 2010, 2012, 2014, 2014, 2018 and 2020. The SOC and TN contents were determined by combustion of air-dried soils using a Vario EL III Elemental Analyzer (Elementar). DOC concentration was measured in fresh soils stored at 4 °C following the method of Jones and Willett[74].

Microbial CUE was measured in 2021 in all soils (sampled in 2010, 2012, 2014, 2016, 2018 and 2020) that had been stored at −80 °C using the [18]O-H$_2$O tracer method after 7 days preincubation[54,55] (details below). Analyses of bacterial and fungal diversity were performed on three of the four replicates samples that had been stored at −80 °C in 2021. The stored soil samples (−80 °C) collected in 2010 and 2020 were further subjected to metagenomic sequencing. Fresh soils sampled in 2020 were analyzed for microbial community composition using phospholipid fatty acid (PLFA) analysis according to ref. 75; changes in microbial community composition were presented as molar percentages (mole %) of the PLFA biomarkers for bacteria or fungi. Amino sugars were extracted from air-dried soils and determined following the method of Zhang and Amelung[56] (details below).

## DNA extraction and amplicon sequencing
DNA was extracted from 0.25 g of soil using the PowerSoil Isolation kit (MoBio Laboratories, Carlsbad, CA, USA) according to the manufacturer's instructions. The quality of the purified DNA was assessed based on the 260/280 nm and 260/230 nm absorbance ratios obtained, using a NanoDrop ND-1000 spectrophotometer (NanoDrop Technologies Inc., Wilmington, DE, USA). The DNA was stored at −80 °C until sequencing analysis.

The V4-V5 region of 16S rRNA and the internal transcribed spacer (ITS) region of the rRNA were amplified to construct bacterial and fungal community profiles, respectively, using high-throughput sequencing. Universal primer sets, F515 (5″-GTGCCAGCMGCCGCGC-3′) and R907 (5′-CCGTCAATTCMTTTRAGTTT-3′) targeting the bacterial 16 S rRNA genes and gITS7F (5′-GTGARTCATCGARTCTTTG-3′) and ITS4R (5′-TCCTCCGCTTATTGATATGC-3′) for fungal ITS, were used[76]. The following thermal program was used for the amplification of the 16S rRNA gene: initial denaturation at 95 °C for 3 min, followed by 27 cycles of denaturing at 95 °C for 30 s, annealing at 55 °C for 30 s and extension at 72 °C for 45 s, then a single extension at 72 °C for 10 min and ending at 4 °C. The PCR amplification of the ITS2 rRNA gene was performed as follows: initial denaturation at 94 °C for 5 min, followed by 30 cycles of denaturing at 94 °C for 30 s, annealing at 52 °C

for 30 s and extension at 72 °C for 30 s, and single extension at 72 °C for 10 min and ending at 4 °C. Purified amplicons were pooled in equimolar and paired-end sequenced on an Illumina Nova6000 platform (Illumina, San Diego, USA) according to the standard protocols by Majorbio Bio-Pharm Technology Co. Ltd. (Shanghai, China).

The raw sequences were subjected to quality control with the following criteria: (i) the 300 bp reads were truncated at any site receiving an average quality score of <20 over a 50 bp sliding window, and the truncated reads shorter than 50 bp were discarded, and reads containing ambiguous characters were also discarded; (ii) only overlapping sequences longer than 10 bp were assembled according to their overlapped sequence. The maximum mismatch ratio of the overlap region was 0.2. Reads that could not be assembled were discarded; (iii) samples were distinguished according to the barcode and primers, and the sequence direction was adjusted, exact barcode matching, and 2 nucleotide mismatches in primer matching. Phylotypes (i.e., Amplicon Sequencing Variants, ASVs) were picked using UNOISE3 with default parameters in USEARCH and were identified at the 100% identity level[77]. The taxonomy of each ASV representative sequence was analyzed by the RDP Classifier Bayesian algorithm against the Silva database (https://www.arb-silva.de/) and the UNITE database (https://unite.ut.ee/) using a confidence threshold of 0.7. We used a randomly selected subset of 24,579 and 21,550 sequences per sample for subsequent bacterial and fungal communities' analysis.

## Metagenomic sequencing and data analyses

DNA extract was fragmented to an average size of about 400 bp using Covaris M220 (Gene Company Limited, China) for paired-end library construction. The paired-end library was constructed using NEXTFLEX Rapid DNA-Seq (Bioo Scientific, Austin, TX, USA). Adapters containing the full complement of sequencing primer hybridization sites were ligated to the blunt end of fragments. Paired-end sequencing was performed on Illumina NovaSeq (Illumina Inc., San Diego, CA, USA) at Majorbio Bio-Pharm Technology Co., Ltd. (Shanghai, China).

The paired-end Illumina reads were trimmed of adaptors, and low-quality reads (length <50 bp or with a quality value < 20 or having N bases) were removed by fastp (https://github.com/OpenGene/fastp, version 0.20.0). Metagenomics data were assembled using MEGAHIT[78] (https://github.com/voutcn/megahit, version 1.1.2). Contigs with a length ≥300 bp were selected as the final assembling result, and then the contigs were used for further gene prediction and annotation. Representative sequences of non-redundant genes were aligned to the KEGG database[79] (http://www.diamondsearch.org/index.php, version 0.8.35) against the Kyoto Encyclopedia of Genes and Genomes database (http://www.genome.jp/keeg/). Carbohydrate-active enzyme annotation was conducted using hmmscan (http://hmmer.janelia.org/search/hmmscan) against the CAZy database (http://www.cazy.org/). We used the RPM (reads per million) method to normalize the relative abundance of pathways and genes annotated within the KEGG and CAZy database[80].

## Microbial carbon use efficiency

Microbial CUE was measured using the $^{18}O$-$H_2O$ tracer method[54,55] on soils previously stored at −80 °C according to ref. 81. Soil samples were adjusted to 40% field water capacity with Millipore water (replenished according to weight loss every 3 days) and pre-incubated for 7 days at 25 °C in tubes covered by vented sealing films to avoid excessive $CO_2$ accumulation. After the pre-incubation, 1 g pre-incubated soil was transferred to two 2 mL Falcon tubes (0.5 g soil in each tube). One sample was amended with $^{18}O$-enriched water to achieve 30 atom% $^{18}O$-labeled soil water, and the other was amended with the same volume of non-labeled Millipore water as a natural abundance control. All samples were then incubated for 48 h at 20 °C in the dark at 60% field water capacity. The $CO_2$ produced during this time was measured using an Agilent 7890 A gas chromatograph (Agilent Technologies, Aito Palo, CA) equipped with a TCD detector. Soil DNA was extracted using the

PowerSoil Isolation kit (MoBio Laboratories, Carlsbad, CA, USA). The extracted DNA (50 μL) was dried in silver capsules at 60 °C for 2 days. Subsequently, the $^{18}O$ abundance and total O content were determined using an elemental analyzer (FLASH 2000, Thermo Fisher Scientific, Cambridge, UK) coupled with an isotope ratio mass spectrometer (IRMS) system (ConFlo VI interface and MAT253 IRMS, Thermo Scientific, Bremen, Germany) at the Chinese Academy of Sciences Institute of Subtropical Agriculture, (Changsha, Hunan Province, China).

Microbial CUE was calculated based on the following equation:

$$CUE = \frac{C_{Growth}}{C_{Growth} + C_{Respiration}} \tag{1}$$

## Microbial necromass C analysis

Glucosamine, galactosamine and muramic acid were used as biomarkers for microbial residues ("necromass"). They were extracted and determined following the method of Zhang and Amelung[56]. Air-dried soil samples (<0.25 mm) were hydrolyzed with 6 M HCl at 105 °C for 8 h after adding 100 μl myo-inositol (internal standard). The derivatized compounds were separated on a gas chromatograph equipped with an HP-5 column and quantified using a flame ionization detector (Agilent 6890 A, Agilent Technologies, Littleton, CO, USA). Bacterial and fungal residue (necromass) C were calculated based on the following equations:

$$\text{Bacterial residue C} = \text{muramic acid} \times 45 \tag{2}$$

$$\text{Fungal residue C} = (\text{mmol glucosamine} − 2 \times \text{mmol muramic acid}) \times 179.2 \times 9 \tag{3}$$

where 45 is the conversion value to the bacterial residue; 179.2 is the molecular weight of glucosamine; and 9 is the conversion factor of fungal glucosamine to the fungal residue. The total microbial residue was estimated as the sum of fungal and bacterial residues.

## Root biomass and exudation C input

Root biomass and root exudation C input were measured from October 2019 to May 2020 during winter wheat growth. During the elongation, flowering and maturation stages of wheat, we collected root exudates from six individual plants in four subplots of each treatment. We used a modified culture-based cuvette system developed especially for root exudates in situ field collection[52,53]. Firstly, wheat roots with the same height and growth status were selected, and terminal fine roots were carefully excavated from the topsoil (0–15 cm) by hand and extensively washed to remove adhering soil particles. Secondly, the intact roots were placed into cuvettes, and then cuvettes were filled with glass beads (c. 1 mm diameter), which were placed to simulate soil particles, and sealed with a special rubber septum. Thirdly, the cuvettes (including the controls with beads only) were covered in foil and reburied in the excavated area in soil. After a one-day equilibration period, a small amount of fresh C-free nutrient solution (0.5 mM $NH_4NO_3$, 0.1 mM $KH_2PO_4$, 0.2 mM $K_2SO_4$, 0.2 mM $MgSO_4$, 0.3 mM $CaCl_2$) was flushed through each cuvette to remove soluble C. Finally, the exudates were collected by flushing the cuvette three times with 15 ml of the fresh nutrient solution after the 24-h incubation. The control cuvettes filled with only coarse silica sand were similarly placed and processed at each plot. The trap solutions were filtered through sterile 0.22 μm syringe filters within 2–5 h of collection and then stored at −20 °C until further analysis. Exudates were collected for three consecutive days. Total non-particulate organic C accumulated in the trap solutions was analyzed using a Multi 3100 N/C TOC analyzer (Analytik, Jena, Germany).

Root exudation rates for each treatment were calculated by subtracting the C accumulation within the root-filled cuvettes from the C

detected in the root-free control and were then expressed as mg C g$^{-1}$ root biomass C h$^{-1}$. We estimated the whole growth period root exudation rates by weighting root exudation rates in each growth period. We calculated yearly root exudates by multiplying the growth period living root biomass with the weighted root exudation rate and the number of hours per growth period[82].

The following equation to assess the whole growth period C exudation of wheat root (c)

$$c = \frac{\Delta c * V}{G * T} * S * 24 * d \quad (4)$$

where $\Delta c$ is the TOC concentration difference between root exudate solution and blank control (mg·L$^{-1}$); $V$ is the volume of root exudate solution (L); $G$ is the root weight (g); $T$ is the collection time (h); $S$ is root biomass (g C m$^{-2}$); 24 is 24 h per day; and $d$ represents the growth period.

## Statistical analyses

Mixed models for repeated measures analysis of variance (ANOVAs) were used to examine the effects of management, warming, year and their interactions on SOC, bacteria and fungi diversity, microbial physiology and necromass. In addition, multiple comparisons (LSD) were used to examine the differences in different parameters among treatments. Linear regressions were used to detect linear trends of SOC, microbial physiological traits and necromass C with time under four treatments. The slopes of those linear relationships were analyzed and compared by Standardized Major Axis regression analysis using the *smatr* package in R v.3.2.1[83]. Cohen's $d$ was further calculated as an estimate of multiple-treatment effect sizes on SOC, microbial physiology and necromass values' response to warming under conservation and conventional agriculture by comparing them against the no warming control; positive $d$ values indicate that the response variables in the treatment have a larger value than in the control, and vice versa[57]. The effect size analyses were performed in the R software v.3.2.1 with the package *effsize*.

Principal component analysis was used to assess changes in the bacterial and fungal communities. Three non-parametric multivariate statistical tests: nonparametric multivariate analysis of variance (ADONIS), analysis of similarity (ANOSIM) and multi-response permutation procedure (MRPP), were used to test the differences in soil microbial communities under warming and control treatments. These analyses were performed in R v.3.2.1 with the *vegan* package. The influence of treatments on the relative abundance of bacterial and fungal taxa was evaluated based on linear mixed-effect models following the methods of ref. 84. Statistical significance is based on Wald type II $\chi^2$ tests and all estimated effect sizes ($\beta$) are based on rescaled response variables. These analyses were performed in R v.3.2.1 with the *lme4* and *car* packages.

The time-decay relationships (TDR) of bacterial and fungal communities were evaluated using the linear regression between logarithmic β similarities and logarithmic temporal distance. The moving window approach was used to assess time decay in microbial communities as previously described[57], which is currently the dominant approach for TDRs. In our biennial surveyed bacterial and fungal diversity data of six sampling dates over 10 years, subset window 1 included the pairwise similarity of samples that were two years apart; subset window 2 is the pairwise similarity of samples four years apart, and so on. So, in this ten-year record, there are 5 two-year intervals, 4 four-year intervals, 3 six-year intervals, 2 eight-year intervals, and 1 ten-year intervals for each plot. Considering the repeated measures design, TDR analysis counted only pairwise comparisons among time points within each plot (that is, 15 pairwise comparisons for each plot and a total of 45 pairwise comparisons for each treatment). We further evaluated the impact of warming on the succession of soil bacterial and fungal communities using the distance of microbial communities between warming and

control at each block in each year for conservation and conventional agriculture, respectively following the method of ref. 57.

The response ratio of carbon metabolism-associated functions of microorganisms between warming and un-warming plots at conservation and conventional agriculture was calculated according to Curtis and Wang[85]. The means of the treatment ($\overline{X}_t$) and control group ($\overline{X}_c$) were used to compute a response ratio using:

$$RR = \ln(\overline{X}_t / \overline{X}_c) = \ln(\overline{X}_t) - \ln(\overline{X}_c) \quad (5)$$

The natural log was used for statistical tests. If $\overline{X}_t$ and $\overline{X}_c$ are normally distributed and both are greater than zero, $\ln(\overline{X}_t / \overline{X}_c)$ is approximately normally distributed with a mean equal to the true response ratio.

SEM analysis was applied to investigate the direct and indirect effects of soil microclimate (soil temperature and moisture), aboveground biomass, DOC and microbial variables (microbial alpha diversity, community structure, microbial CUE and necromass) on SOC. We first considered a hypothesized conceptual model (Supplementary Fig. 10) that included all reasonable pathways. Temperature was only selected to improve model fit because we observed strong collinearity between soil temperature and moisture (Pearson's $R^2 > 0.78$). We sequentially eliminated non-significant pathways unless the pathways were biologically informative, or added pathways based on the residual correlations. A total of 72 independent soil samples were used to run the SEM. The overall goodness of fit of the SEM results was evaluated by the Chi-square test ($\chi^2$) and the goodness fit index (GFI). When $0 \le \chi^2/df \le 2$ and GFI > 0.9, the model has a good fit. The SEM analysis was performed using AMOS 23.0 (AMOS Development Corporation).

## Reporting summary

Further information on research design is available in the Nature Portfolio Reporting Summary linked to this article.

## Data availability

The DNA sequences of the 16S rRNA gene and ITS amplicons in this study have been deposited in the National Center for Biotechnology Information (NCBI) under project accession numbers PRJNA903096 and PRJNA903090. Raw shotgun metagenomic sequences in this study have been deposited in the National Center for Biotechnology Information (NCBI) under project accession PRJNA1007786. Silva database is available at https://www.arb-silva.de/. UNITE database is available at https://unite.ut.ee/. Source data are provided in this paper. Source data are provided with this paper.

## Code availability

The analysis code that supports the findings of this study is available at GitHub https://github.com/bio-carbon/code.

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

## Acknowledgements

This study was supported by the the National Natural Science Foundation of China (grant nos. 32071629, 32071607), National Key R&D Program of China (2022YFD1901300), 2115 Talent Development Program of China Agricultural University and Beijing Advanced Disciplines and Strategic Priority Research Program of the Chinese Academy of Sciences (XDA28130301). The manuscript revised by Y.K. was also supported by the RUDN University Strategic Academic Leadership Program.

## Author contributions

All authors contributed intellectual input and assistance to this study and manuscript preparation. J.T., J.A.J.D., F.Z., M.F.C., and J.Z. designed the original concept and experiment strategy. J.T. performed the lab experiments and collected the data. R.H. carried out the field experiment and collected the data. J.T. analyzed the data with help of Y.D. J.T. drafted the manuscript with help from J.A.J.D., Y.Y., I.P.H., M.F.C., Y.K. and J.Z. All authors contributed to the article and approved the submitted versions.

## Competing interests

The authors declare no competing interests.
