## [Peer Review File · Nature Communications]

Microbially mediated mechanisms underlie soil carbon accrual by conservation agriculture under decade-long warmingREVIEWER COMMENTS

Reviewer #1 (Remarks to the Author):

The article of Tian and co-authors is asking a relevant and timely question which is: how carbon storing soils respond to climate warming? To answer this question the authors sampled a 10 years old tillage and warming experiment to evaluate how conservation agriculture (no tillage) and traditional tillage impact soil carbon cycling. Their results can help us understand mechanistically how agricultural management linked to the direct and indirect influences of warming can impact soil carbon cycling. I think this paper is of high interest for the broad audience of Nat Comm. However, I would like to make some suggestions to the authors regarding the presentation of their results. I consider that this article contains an interesting dataset to disentangle between direct and indirect impact of warming in agricultural soils and its consequence for SOC. I would suggest authors to more clearly state that already at the introduction, right now they only mention in one sentence the impact of warming on moisture (L116-118). This is a very important aspect to understand this dataset so I would suggest to phrase it more clearly from the beginning and throughout the manuscript. The main take-home message of their results is that warming can result in a higher plant biomass production and if this biomass stays in the soil it can promote microbial growth efficiency and SOC retention in the long-term. However, when reading the paper this main message doesn't appear to be very clear.

Right now the results are presented very individually (e.g. warming decreases variable A, warming increases variable B, etc). It would be helpful for the reader if authors present their results in a more integrated format. The authors should also mention some aspects of the methods that are important to understand the results (as in Nat. Comm. the methods are only at the end of the manuscript). Some methodological aspects that need to be taken into consideration to understand the results: the warming is induced via infrared heaters (+2C) suspended on 3m above the ground. Another important aspect in my opinion is that in the conservation agriculture all "plant residues were chopped to approximately 5 cm in length and retained on the soil surface" while in the traditional tillage treatment "the residues were removed after the maize harvest". This implies that the bare soil of traditional tillage is more directly exposed to the warming induced by the infrared heaters, while in the conservation agriculture the soil might be more "protected" to the radiation due to the plant biomass in the soil. This could help explain why higher average temperatures are observed in the traditional tillage compared to the conservation tillage (L169-170), as well as why the decrease in soil moisture with warming was higher in traditional compared to conservation tillage (L171-172). These indirect influences of warming on the soil environment (moisture) can help explain the differences regarding the above ground plant biomass. As plants are limited by water they can't growth equally on both treatments. Further, the above ground plant residues are left on top of the soil and can be slowly incorporated into the soil during decomposition under conservation agriculture. Other results supporting this line of thinking is their PLFA results and the fungal/bacterial PLFAs. There is substantial evidence suggesting that fungi is a dominant driver of plant biomass decomposition. The very strong differences observed in the PLFAs between traditional and conservation tillage suggest that plant input into the system is influencing the abundance of microorganisms (MBC) and is the major driver of SOC dynamics in these agricultural soils. This is a very important result that authors are not highlighting in their paper. Authors do suggest that there are interaction effects between warming and management on SOC (L156-161: Along with the main effects of management and warming as individual treatments, there were interactive effects on SOC content of management \times warming and management \times warming \times year ($p < 0.05$; Supplementary Table 1), indicating that a positive stimulatory effect of warming on SOC was modulated by management and that this interaction varied with time.) However, while they use the term "management" they could more directly mention the relevance of plant input as the driver of SOC dynamics. This is not only a crucial aspect in this study, it is very important to contextualize their results.

L456-459: here authors could also include the differences in plant input as a consequence of different tillage management.

L467-468: "Warming increased the total and fungal necromass carbon accumulation under conservation agriculture" or was the increased plant biomass due to warming and therefore,

warming had an indirect effect?

L481-483: Again, this is only possible due to higher nutrient inputs (plant biomass) in the warming treatment. Otherwise, it has been widely reported that increasing temperatures result in a decrease in CUE because respiration responds stronger than growth to increases in temperature.

L488-508: In their concluding paragraph I think it is very important for authors to contextualize their results. Right now when reading it we have the impression that conservation agriculture can deal with warming sequestering more carbon. But this main message is missing the main point of these results which highlight the complex interplay between warming, increased plant biomass and changes in soil moisture controlling SOC dynamics. Global change is not only increasing temperature but also changing precipitation patterns. It is important that authors bring their results into the appropriate context. If decreased precipitation impedes the expected increase in production of plant biomass due to warming (as shown in their experiment), there cannot be a positive impact of warming on SOC.

I find that these results are very interesting and valuable for the field representing 10 years of data collection. Once authors address these comments it will be certainly an important paper in a growing field of research and of high interest for the broad readership of *Nature Communications*'s journal.

Other comments:

Methods: it is not clear how many times authors measured CUE. Were measurements taken in 2010, 2014, 2016 and 2020? Please clarify this in the text and in the figure legends (including supplementary figure legends). Please clarify for every measurement/assay when the measurement was performed, the methods section is not very clear. Authors wrote in L955: "In our biennial survey data, subset window 1 included the pairwise similarity of samples that were two years apart; subset window 2 is the pairwise similarity of samples four years apart, and so on. In this ten-year record, there are 5 two-years intervals, 4 four-years intervals," ... But for which datasets? Please clarify, one suggestion would be to create a table to show which data was collected at each year.

SEM analysis: how many data points were used in the SEM? Did authors compile all the multi-year datasets into the analysis? In which years the bacterial and fungal communities were sequenced?

Figure 2: authors could also show their microbial biomass carbon (MBC) measurements in this figure as an extra panel.

L160: "Warming progressively increased SOC content at an average rate of 0.7% per year from 2010 to 2020 under conservation agriculture)" Phrasing the results like this is dangerous. Most likely this is caused by the increased plant inputs due to warming in combination with conservation agriculture. Normally in the literature warming is associated with decreased SOC, so please rephrase your sentence taking into account the full experimental design.

L85: the use of the term temporal scales here is unclear

L104-105: there is a recent article with some interesting results that authors could introduce here: Kpemoua et al., 2023, *Soil Biology and Biochemistry*. Are carbon-storing soils more sensitive to climate change? A laboratory evaluation for agricultural temperate soils, <https://doi.org/10.1016/j.soilbio.2023.109043>

L106-114: Here authors are mentioning some references that are already 10 years old. These are relevant references in this field but it would be nice to include also some of the most recent papers evaluating the question of microbial acclimation to warming. Here are examples of some more recent relevant papers:

Alster, C. J., Robinson, J. M., Arcus, V. L., & Schipper, L. A. 2022. Assessing thermal acclimation of soil microbial respiration using macromolecular rate theory. *Biogeochemistry*, 158, 1– 11.
Domeignoz-Horta, L. A., et al., 2023. Substrate availability and not thermal acclimation controls

microbial temperature sensitivity response to long-term warming. *Global Change Biology*, 29, 1574– 1590. <https://doi.org/10.1111/gcb.16544>

L116-118: It is interesting to mention how warming can impact other properties that influence microbial physiology and therefore indirectly impact physiology. However, in their introduction as authors did not study moisture I would suggest to restrict their introduction to warming as direct driver on physiology. A recent and interesting paper on this aspect that would fit well in this section is: Söllinger, A., et al., (2022). Down-regulation of the bacterial protein biosynthesis machinery in response to weeks, years, and decades of soil warming. *Science Advances*, 8(12), eabm3230. <https://doi.org/10.1126/sciadv.abm3230>

L135-138: Is this experiment also a warming experiment? From this description we can't understand if it is a warming and tillage experiment. Please rephrase this section to clarify. In the last paragraph of the introduction it is important that authors concisely summarize their methods so that authors can understand their results section.

L812: five or three times?

L384: replace was by is required.

Data availability statement: authors could make available in the supplementary files some files with their raw dataset. Right now authors write "The authors declare that the data supporting the findings of this study are availability with the article and its supplementary Information files, and from the corresponding authors on request." However, none of their supplementary files contain their raw data.

Reviewer #2 (Remarks to the Author):

The article "Microbially mediated mechanisms underlie soil carbon accrual by conservation agriculture under decade-long warming" provides results from a multi-factor field experiment including warming and management treatments. The primary focus of the paper was conventional tillage and its role in accrual of soil organic carbon over the experiment. The authors present evidence of microbial community shifts, particularly the fungal community, that may be responsible for the increase in SOC. The heavy focus on microbial measures allows for a more mechanistic understanding of why SOC accrual is occurring under conservation tillage even with warming. This helps improve the understanding of SOC formation and stabilization in soils, and highlights the role of fungi in this accumulation.

The methodology used in this manuscript are appropriate. In particular, the use of isotopes for CUE improves the accuracy of the CUE estimates. The results support the interpretation and conclusion of the work. Figure 6 is very helpful for following the conclusions

One suggestion in the results: The results text initially follows a pattern where conservation tillage data is presented and then traditional tillage. Then it switches to traditional followed by conservation. It might easier for the reader to follow if the same order was maintained throughout the results.

Reviewer #3 (Remarks to the Author):

The authors studied the effect of warming in combination with 'conservation agriculture' on Soil Organic Carbon (SOC) dynamics in an agricultural soil in northern China. They found that accelerated fungal succession and microbial community-level physiological adjustment towards high growth efficiencies and metabolic byproducts under conservation agriculture promoted SOC formation and accrual in warmed conditions over 10 years.

The topic of this research is important and of great interest to the readers. As global climate change is rapidly advancing, and the earth is warming, we will increasingly require agricultural practices to adapt so that more carbon can be stored in soils. I also find that the experimental design of the experiment is sound and that the manuscript contains extensive amounts of data, including measurements of dissolved organic carbon, root biomass and root exudation carbon, aboveground biomass indexes, phospholipid fatty acid analysis, bacterial and fungal rRNA gene amplicon sequencing, glucosamine, galactosamine and muramic acid analysis to characterize necromass, metagenomic sequencing, and microbial carbon use efficiency measurements using the ^{18}O - H_2O tracer method, which also provides measures of microbial growth. This is a rich and important data set.

However, I don't find the manuscript to be ready for publication because the writing is not precise and it is often unclear what argument the authors are making. Let me illustrate this by considering the abstract of the paper. I focus only on the abstract because it is an important component of a research paper that should summarize the findings of the paper and because I don't have the time nor resources to rewrite the entire paper. But the entire paper not just the abstract needs to be rewritten.

L53 Line 53 mentions "conservation agriculture." Define more clearly what conservation agriculture is. Is it simply "no tillage"? If so why not call it that?

L53 "A positive warming effect on SOC strengthened under conservation agriculture" It is unclear what a positive warming effect is. Does this mean that warming increased SOC concentrations?

L54 Warming accelerated fungal community turnover under conservation agriculture, as indicated by high temporal scaling rates and increasingly divergent succession." What are high temporal scaling rates and divergent succession? Does this mean there was an increase in fungal diversity under the warming treatment?

L59 "In parallel, warming increased microbial growth-related genes". It is unclear to me how warming can increase genes. Does this mean that the relative abundance of growth related genes increased? Or is it absolute abundance? Or did the diversity of growth related genes increase?

L63 microbial community-level physiological adjustment towards high growth efficiencies and metabolic byproducts". What is a community level physiological adjustment? And how can it be adjusted towards metabolic by products?

In my opinion before the manuscript can be properly reviewed it is important that the writing becomes more precise so that its meaning is unambiguous and not left to the interpretation of the reader.

Response to Reviewers' comments

A. Response to Reviewer #1 (Remarks to the Author):

A1. *The article of Tian and co-authors is asking a relevant and timely question which is: how carbon storing soils respond to climate warming? To answer this question the authors sampled a 10 years old tillage and warming experiment to evaluate how conservation agriculture (no tillage) and traditional tillage impact soil carbon cycling. Their results can help us understand mechanistically how agricultural management linked to the direct and indirect influences of warming can impact soil carbon cycling. I think this paper is of high interest for the broad audience of Nat Comm.*

Response: We appreciate the Reviewer for their supportive comments and for highlighting the strengths and the novelty of our study.

A2. *However, I would like to make some suggestions to the authors regarding the presentation of their results. I consider that this article contains an interesting dataset to disentangle between direct and indirect impact of warming in agricultural soils and its consequence for SOC. I would suggest authors to more clearly state that already at the introduction, right now they only mention in one sentence the impact of warming on moisture (L116-118). This is a very important aspect to understand this dataset so I would suggest to phrase it more clearly from the beginning and throughout the manuscript.*

Response: We thank the Reviewer for their suggestions to emphasize the importance of soil moisture. We have expanded the Introduction by providing an improved overview of abiotic and biotic factors, as the Reviewer suggested:

Lines 112-122 “Microbial CUE depends on both abiotic and biotic factors²³, which are affected by warming and management practices. Increased temperature can have both direct and indirect effects on microbial CUE. Generally, warming decreases microbial CUE, as a greater proportion of the substrate is reallocated from growth to maintenance metabolism^{19,27}, which alters rates of enzyme-driven processes²⁸⁻³⁰. Warming can alter CUE indirectly via changes in soil moisture, substrate availability or the composition and/or structure of microbial communities^{27,31}. Warming may decrease soil moisture and reduce microbial CUE because more substrate is allocated to dissimilatory metabolism, and hence less available for growth^{19,32}. In contrast, warming may enhance plant growth, productivity and rhizosphere carbon input^{33,34}, promoting microbial growth and necromass accumulation³⁵.”

We have also extended the description of the Results to incorporate more information about the effects of warming on soil moisture:

- Lines 189-199 “The conservation agriculture treatment, characterized by continuous soil cover by crop residues with no tillage, increased SOC, which mitigated the effects of experimental warming on soil temperature and moisture contents. Over the 10-year study period, we continuously monitored soil temperature and moisture using in-field sensors. We observed that both of these factors were changed by management and experimental warming. As expected, experimental warming increased soil temperature but decreased soil moisture under both conservation and conventional agriculture ($p < 0.05$; Figs. 1b and c). However, the warming effects were modulated by management. Soils covered by crop residues under conservation agriculture were cooler (1.5°C vs. 1.9°C ; $p < 0.05$; Fig. 1b) and wetter (8.9 % vs. 11%, $p < 0.05$; Fig. 1c) compared with conventional agriculture.”

- Lines 205-208 “Warming increased belowground plant carbon inputs, including root biomass and root exudate C, in both conservation and conventional agriculture ($p < 0.05$; Figs. 1d 1e and 1f), presumably in response to soil drying under warm conditions.”
- Lines 381-384 “A total of 69% of the SOC variations were accounted for by the combinations of soil temperature and moisture, substrate availability (indicated by plant carbon inputs and DOC), microbial alpha diversity, microbial community composition, microbial CUE and necromass carbon (Fig. 5a).”

In the Discussion, we have elaborated on the management effect on soil moisture:

- Lines 422-425 “Herein, SOC content increased exclusively under conservation agriculture and was further boosted by 10 years of warming (Figs. 1a and j), which partially can be attributed to the continuous soil cover that mediated the effects of warming on soil temperature and prevented excessive drying.”
- Line 493-496 “This group is characterized by oligotrophic attributes, meaning they thrive in low-nutrient environments⁶⁸ and are closely associated with decreases in net nitrogen mineralization due to the reduced substrate availability caused by soil drying⁶⁶.”
- Lines 496-498 “Therefore, the slow temporal turnover of bacterial communities may be attributed to a large portion of inactive (dormant or slow growing) bacteria in warmer and drier soils⁶⁹.”

A3. *The main take-home message of their results is that warming can result in a higher plant biomass production and if this biomass stays in the soil it can promote microbial growth efficiency and SOC retention in the long-term. However, when reading the paper this main message doesn't appear to be very clear. Right now the results are presented very individually (e.g. warming decreases variable A, warming increases variable B, etc). It would be helpful for the reader if authors present their results in a more integrated format.*

Response: We thank the Reviewer for advising on the presentation of our manuscript. As advised, we have reviewed and made multiple changes in the Results section to provide a more integrated and cohesive narrative to give better context to our Results. There are too many changes to give all examples in the response letter, and we hope that the Reviewer has time to read our Results section again and hope that the changes meet the Reviewer's satisfaction.

A4. *The authors should also mention some aspects of the methods that are important to understand the results (as in Nat. Comm. the methods are only at the end of the manuscript). Some methodological aspects that need to be taken into consideration to understand the results: the warming is induced via infrared heaters (+2C) suspended on 3m above the ground.*

Response: We appreciate the Reviewer's comment. We have incorporated key aspects of the method throughout this section, including a new Figure describing the field experiment in the Supplementary materials (Fig.S1), for example:

In the Introduction:

Lines 159-163 “To test these hypotheses, we measured SOC, bacterial and fungal communities using DNA sequencing, microbial functions using metagenomics, and microbial physiological traits (CUE and microbial necromass) using substrate independent H₂¹⁸O labeling and microbial biomarker (amino sugars) analysis.

In the Results:

- Lines 169-171 “Warming the field plots under conservation or conventional agriculture was experimentally imposed for 10 years using infrared heaters, maintaining the soil temperature at +2°C above ambient levels (Supplementary Fig.1).”
- Line 191-193 “Over the 10-year study period, we continuously monitored soil temperature and moisture using in-field sensors.”
- Lines 216-217 “To address it, we measured soil microbial community CUE using the substrate-independent ¹⁸O-H₂O method^{54,55}.”
- Lines 245-248 “The acceleration of microbial turnover in response to warming under conservation agriculture was substantiated by a 77% increase in total microbial necromass carbon (indicated by the concentration of amino sugar biomarker⁵⁶) compared with un-warmed soils ($p < 0.05$; Fig. 2c; Supplementary Table 2).”
- Lines 272-274 “To understand the role of the soil microbiome in driving SOC dynamics in response to warming and management, we examined how microbial diversity and community turnover co-varied over the 10 years using DNA sequencing.”
- Lines 341-342 “We further used shotgun metagenomic analysis to determine the changes in microbial functions related to cellular processes and metabolism (Fig. 4a).”

A5. *Another important aspect in my opinion is that in the conservation agriculture all “plant residues were chopped to approximately 5 cm in length and retained on the soil surface” while in the traditional tillage treatment “the residues were removed after the maize harvest”. This implies that the bare soil of traditional tillage is more directly exposed to the warming induced by the infrared heaters, while in the conservation agriculture the soil might be more “protected” to the radiation due to the plant biomass in the soil. This could help explain why higher average temperatures are observed in the traditional tillage compared to the conservation tillage (L169-170), as well as why the decrease in soil moisture with warming was higher in traditional compared to conservation tillage (L171-172). These indirect influences of warming on the soil environment (moisture) can help explain the differences regarding the above ground plant biomass. As plants are limited by water they can’t grow equally on both treatments. Further, the above ground plant residues are left on top of the soil and can be slowly incorporated into the soil during decomposition under conservation agriculture.*

Response: We thank the Reviewer for making this important point. We refer the Reviewer to our response A2 above. We have followed the suggestions to emphasize the importance of soil moisture with reference to the combined effect of crop residues and warming.

A6. *Other results supporting this line of thinking is their PLFA results and the fungal/bacterial PLFAs. There is substantial evidence suggesting that fungi is a dominant driver of plant biomass decomposition. The very strong differences observed in the PLFAs between traditional and conservation tillage suggest that plant input into the system is influencing the abundance of microorganisms (MBC) and is the major driver of SOC dynamics in these agricultural soils. This is a very important result that authors are not highlighting in their paper.*

Response: We thank the Reviewer for drawing attention to our PLFA data. We agree with the Reviewer that the plant C input is an important driver that influences the fungal biomass and necromass accumulation. We have revised and updated the Discussion section based on the Reviewer’s suggestions.

- Lines 256-262 “An increase in the concentration of microbial PLFA biomarkers, which serve as indicators of microbial biomass of dominant groups (Fig. 1h), also indicated the benefits of conservation management and warming for fungi, as reflected in larger fungal/bacterial PLFA ratios

(Fig. 1i). The larger fungal biomass consequently led to larger fungal necromass, contributing significantly to total necromass over time ($R^2 = 0.16-0.62$, $p < 0.05$; Supplementary Fig. 4) and was further increased by warming (by 29%; $p < 0.05$; Fig. 2c)."

- Lines 439-447 "Root biomass and root exudation are major forms of carbon input into soils, which serve as important carbon sources for soil microorganisms. Together with crop residue retention, DOC, aboveground biomass, belowground root biomass and rhizodeposition (Figs. 1d-1g) increased under long-term conservation agriculture with warming, indicating higher plant carbon inputs (Figs. 1d-f) that provided abundant substrates for microbial proliferation. This particularly favored fungal growth, leading to higher fungal biomass (Fig. 1h) and fungal/bacterial ratios (Fig. 1i) under long-term conservation management with warming."

A7. *Authors do suggest that there are interaction effects between warming and management on SOC (L156-161: Along with the main effects of management and warming as individual treatments, there were interactive effects on SOC content of management \times warming and management \times warming \times year ($p < 0.05$; Supplementary Table 1), indicating that a positive stimulatory effect of warming on SOC was modulated by management and that this interaction varied with time.) However, while they use the term "management" they could more directly mention the relevance of plant input as the driver of SOC dynamics. This is not only a crucial aspect in this study, it is very important to contextualize their results.*

Response: We thank the Reviewer for their recommendation to emphasize the key driver of SOC and microbial responses. We have revised the manuscript to draw attention to this important point, as below:

In the Abstract:

- Lines 57-58 "These increases arose from raised plant carbon input, which indirectly controlled microbial CUE via changes in fungal communities."

In the Introduction:

- Lines 122-124. "In contrast, warming may enhance plant growth, productivity and rhizosphere carbon input^{33,34}, promoting microbial growth and necromass accumulation³⁵."
- Lines 151-154 "We hypothesized that, under conditions of climate warming, (i) conservation agriculture increases SOC directly through increased plant-derived carbon inputs and indirectly via greater substrate availability to the soil microbial community."

In the Results

- Lines 167-168 "Conservation agriculture increased SOC by mediating the effect of warming on soil and plant properties."
- Lines 204-215 "We also evaluated plant carbon input using gravimetric measurements of aboveground biomass and root biomass, and root exudation carbon using a method developed for *in situ* collection of roots exudates^{52,53}. Warming increased aboveground biomass under conservation agriculture, but not under conventional agriculture ($p < 0.05$; Fig. 1d), probably due in part to the positive effects of residue retention and no tillage on soil moisture. Warming increased belowground plant carbon inputs, including root biomass and root exudate carbon, in both conservation and conventional agriculture ($p < 0.05$; Figs. 1d 1e and 1f), presumably in response to soil drying under warm conditions. Overall, total root carbon input increased by 65% under warming compared with ambient control under conservation agriculture ($p < 0.05$; Supplementary Fig. 2), ultimately contributing to the increase in SOC (Fig. 1a) and DOC (Fig. 1g) concentrations."
- Lines 337-341 "Management and warming changed the magnitudes of plant carbon inputs, including crop residues from aboveground biomass, root biomass and root exudates (Fig. 1). These

changes had cascading effects on the biomass, activity (Figs. 1h and 1i; Fig. 2), structure and turnover (Fig.3) of the soil microbial communities, particularly the proliferation of fungi.”

- Lines 357-361 “Changes in the genes encoding carbohydrate-active enzymes based on the CAZy database (Fig. 4b) provided insights into how management and warming affected plant carbon inputs, including lignin, cellulose and pectin from plant cell walls, and microbial inputs such as chitin from fungal biomass (indicated by the increase in fungal glucosamine; Fig. 2c).”

In the Discussion

- Lines 441-445 “Together with crop residue retention, DOC, aboveground biomass, belowground root biomass and rhizodeposition (Figs. 1d-1g) increased under long-term conservation agriculture with warming, indicating higher plant carbon inputs (Figs.1d-f) that provided abundant substrates for microbial proliferation.”
- Lines 510-512 “Microbial necromass accumulation in soil is driven by plant carbon input and mediated by microbial necromass production and stabilization³⁵.”
- Lines 532-535 “In particular, under long-term conservation agriculture with warming, the increase in plant carbon input accelerated fungal succession and enhanced microbial growth efficiencies, leading to a progressive increase of microbially-derived carbon contributions to SOC formation and accrual at decadal timescales.”

A8. L456-459: here authors could also include the differences in plant input as a consequence of different tillage management.

Response: We thank the Reviewer for this important recommendation. We have made the following revisions:

- Lines 478-482 “Fungi have a greater carbon demand than bacteria and are usually the first to utilize belowground plant carbon inputs⁶². Consequently, the rapid fungal growth and fungal community changes, driven by increased plant carbon input, are key drivers of temporal community turnovers under conservation agriculture with warming.”
- Lines 498-501 “These results suggest that microbial physiological adjustments that selected microbial lineages were better adapted to changing soil microclimatic conditions and substrate availability over time.”

A9. L467-468: “Warming increased the total and fungal necromass carbon accumulation under conservation agriculture” or was the increased plant biomass due to warming and therefore, warming had an indirect effect?

Response: We appreciate the Reviewer’s comments. We have followed the suggestion by adding discussions in L508-517.

Lines 510-517 “Microbial necromass accumulation in soil is driven by plant carbon input and mediated by microbial necromass production and stabilization³⁵. Accordingly, the fungal community was positively correlated with microbial CUE, which had the largest positive effects on fungal necromass (Figs. 5a and 5b). These results suggested that warming increased microbial growth efficiency and fungal community turnover due to greater plant carbon input, which ultimately increased the supply of microbial residues contributing to SOC formation.”

A10. L481-483: Again, this is only possible due to higher nutrient inputs (plant biomass) in the warming treatment. Otherwise, it has been widely reported that increasing temperatures result in a decrease in CUE because respiration responds stronger than growth to increases in temperature.

Response: We appreciate the Reviewer's comments. We have improved this sentence according to Reviewer's comment.

Lines 520-523. "Overall, our results suggested that warming enhances SOC formation and accumulation under conservation agriculture by increasing microbial growth efficiency via accelerating fungal community turnovers, as well as via increased fungal necromass carbon accumulation (Fig. 2 and Fig.3)."

A11. L488-508: In their concluding paragraph I think it is very important for authors to contextualize their results. Right now when reading it we have the impression that conservation agriculture can deal with warming sequestering more carbon. But this main message is missing the main point of these results which highlight the complex interplay between warming, increased plant biomass and changes in soil moisture controlling SOC dynamics. Global change is not only increasing temperature but also changing precipitation patters. It is important that authors bring their results into the appropriate context. If decreased precipitation impedes the expected increase in production of plant biomass due to warming (as shown in their experiment), there cannot be a positive impact of warming on SOC.

Response: We thank the Reviewer for the excellent suggestions to improve the summary. We agree with the Reviewer that precipitation interacts with warming, and altered substrates' availability and microclimate conditions play important roles in regulating microbial responses to warming under conservation agriculture. We revised this paragraph according to the Reviewer's comment, as below:

Lines 531-545. "Our research highlights the significant roles of altered substrate availability and microclimate conditions in driving microbial responses to warming under conservation agriculture. In particular, under long-term conservation agriculture with warming, the increase in plant carbon input accelerated fungal succession and enhanced microbial growth efficiencies, leading to a progressive increase of microbially-derived carbon contributions to SOC formation and accrual at decadal timescales. If our findings can be generalized to other systems and regions where water does not limit productivity (e.g., irrigated regions), regenerative management can result in effective carbon sequestration under warming, which increases agricultural resilience as a vital component of 'climate-smart agriculture'. However, more research is needed to understand the combined effects of management and climate on SOC accrual and persistence in arable soils across different regions with a wide precipitation range, which could guide climate-smart land use practices to sequester carbon effectively and optimize crop production in the face of a changing climate."

A12. I find that these results are very interesting and valuable for the field representing 10 years of data collection. Once authors address these comments it will be certainly an important paper in a growing field of research and of high interest for the broad readership of Nature Communication's journal.

Response: We are very thankful to the Reviewer for their positive comments and recommendations to improve our manuscript. We hope that our changes are satisfactory.

Other comments:

A13. Methods: it is not clear how many times authors measured CUE. Were measurements taken in 2010, 2014, 2016 and 2020? Please clarify this in the text and in the figure legends (including supplementary figure legends). Please clarify for every measurement/assay when the measurement was performed, the methods section is not very clear. Authors wrote in L955: "In our biennial survey data, subset window 1 included the pairwise similarity of samples that were two years apart; subset window 2 is the pairwise similarity of samples four years apart, and so on. In this ten-year record, there are 5

two-years intervals, 4 four-years intervals,” ... But for which datasets? Please clarify, one suggestion would be to create a table to show which data was collected at each year.

Response: We apologize for the lack of clarity. In response, we have updated the Method accordingly (Lines 578-597). We also have followed the suggestion and added the following in L750-753.

- Lines 577-593 “Soil samples (0-5 cm depth) were taken using a soil corer (5 cm inner diameter) after winter wheat harvest in 2010, 2012, 2014, 2016, 2018 and 2020. Before laboratory analyses, composited soil samples from each replicate plot were passed through a 2 mm sieve to remove visible roots and gravel. Analysis for soil organic carbon (SOC), dissolved organic carbon (DOC), plant aboveground biomass, microbial biomass, microbial necromass (amino sugars), microbial community CUE, bacterial diversity and fungal diversity, were conducted in 2010, 2012, 2014, 2014, 2018 and 2020 on the samples collected in those years. The SOC and total nitrogen (TN) were determined by combustion using a Vario EL III Elemental Analyzer (Elementar). DOC concentration was measured following the method of Jones and Willett⁷⁴. Measurement of root biomass and root exudation C was only conducted from October 2019 to May 2020 during winter wheat growth according to Phillips *et al.*⁵² (details below). Soils sampled in 2020 were analyzed for microbial community composition using phospholipid fatty acid (PLFA) analysis according to Frostegård *et al.*⁷⁵. Changes in microbial community composition extracted from the soil samples were presented as molar percentages (mole %) of the PLFA biomarkers for bacteria or fungi. Soil samples sampled in 2010 and 2020 were subject to metagenomic sequencing and analysis as below.”
- Lines 748-751 “In our biennial surveyed bacterial and fungal diversity data of six sampling dates over 10 years, subset window 1 included the pairwise similarity of samples that were two years apart; subset window 2 is the pairwise similarity of samples four years apart, and so on.”

A14. SEM analysis: how many data points were used in the SEM? Did authors compile all the multi-year datasets into the analysis? In which years the bacterial and fungal communities were sequenced?

Response: We thank the Reviewer for raising this point and apologize for the lack of clarity. To further investigate the direct and indirect effects of soil microclimate (soil temperature and moisture) and substrate availability (aboveground biomass and DOC) and microbial variables (microbial alpha diversity, community structure, microbial CUE and necromass) on SOC, structural equation modeling (SEM) analysis was performed. All above mentioned parameters in the SEM have six year’s data. We have updated the text as follows:

Lines 768-772 “Structural equation modeling (SEM) analysis was performed by using AMOS 23.0 to further investigate the direct and indirect effects of soil microclimate (soil temperature and moisture) and substrate availability (aboveground biomass and DOC) and microbial variables (microbial alpha diversity, community structure, microbial CUE and necromass) on SOC. All aforementioned parameters in the SEM have six year’s data.”

A15. Figure2: authors could also show their microbial biomass carbon (MBC) measurements in this figure as an extra panel.

Response: We thank the Reviewer for the suggestions. We have provided the MBC data in Supplementary Materials in Fig. 2.

L160: “Warming progressively increased SOC content at an average rate of 0.7% per year from 2010 to 2020 under conservation agriculture)” Phrasing the results like this is dangerous. Most likely this is caused by the increased plant inputs due to warming in combination with conservation agriculture.

Normally in the literature warming is associated with decreased SOC, so please rephrase your sentence taking into account the full experimental design.

Response: We thank the Reviewer for the suggestions. We have deleted the statement from the revised version.

A16. L85: *the use of the term temporal scales here is unclear.*

Response: We apologise to the Reviewer for the lack of clarity. We have deleted the term; the sentence now reads:

Lines 82-83 “Unfortunately, studies investigating the interactive effects of management and warming on SOC accrual in croplands are extremely scarce.”

A17. L104-105: *there is a recent article with some interesting results that authors could introduce here: Kpemoua et al., 2023, Soil Biology and Biochemistry. Are carbon-storing soils more sensitive to climate change? A laboratory evaluation for agricultural temperate soils, <https://doi.org/10.1016/j.soilbio.2023.109043>*

Response: We thank the Reviewer for bringing this useful reference to our attention. We have added the following text with the citation:

Lines 102-104 “However, a short-term 3-month laboratory incubation study detected no significant difference in SOC mineralization between conservation and conventional agriculture under various temperature conditions¹⁸.”

A18. L106-114: *Here authors are mentioning some references that are already 10 years old. These are relevant references in this field but it would be nice to include also some of the most recent papers evaluating the question of microbial acclimation to warming. Here are examples of some more recent relevant papers:*

Alster, C. J., Robinson, J. M., Arcus, V. L., & Schipper, L. A. 2022. Assessing thermal acclimation of soil microbial respiration using macromolecular rate theory. Biogeochemistry, 158, 1– 11.

Domeignoz-Horta, L. A., et al., 2023. Substrate availability and not thermal acclimation controls microbial temperature sensitivity response to long-term warming. Global Change Biology, 29, 1574–1590. <https://doi.org/10.1111/gcb.16544>

Response: We thank the Reviewer for highlighting the need to update our references and for bringing several useful references to our attention. The following references have been added at Lines 114-118. We also added other three recent references in the revised text (Lines 107-122):

- Alster, C. J., et al., (2022). Assessing thermal acclimation of soil microbial respiration using macromolecular rate theory. *Biogeochemistry* 158(1): 131-141.
- Domeignoz-Horta, L. A., et al. (2020). Microbial diversity drives carbon use efficiency in a model soil. *Nature Communications* 11(1): 3684.
- Domeignoz-Horta, L. A., et al., (2023). Substrate availability and not thermal acclimation controls microbial temperature sensitivity response to long-term warming. *Glob Chang Biol* 29(6): 1574-1590.
- Tao, F., et al. (2023). Microbial carbon use efficiency promotes global soil carbon storage. *Nature*.
- Tian, W., et al., (2022). Thermal adaptation occurs in the respiration and growth of widely distributed bacteria. *Glob Chang Biol* 28(8): 2820-2829.
- Wang, N., et al., (2019). Effects of climate warming on carbon fluxes in grasslands- A global meta-analysis. *Glob Chang Biol* 25(5): 1839-1851.

A19. L116-118: *It is interesting to mention how warming can impact other properties that influence microbial physiology and therefore indirectly impact physiology. However, in their introduction as authors did not study moisture I would suggest to restrict their introduction to warming as direct driver on physiology. A recent and interesting paper on this aspect that would fit well in this section is: Söllinger, A., et al., (2022). Down-regulation of the bacterial protein biosynthesis machinery in response to weeks, years, and decades of soil warming. Science Advances, 8(12), eabm3230. <https://doi.org/10.1126/sciadv.abm3230>*

Response: We appreciate the Reviewer's comment. We have followed the suggestions to supplement the Introduction by providing an improved overview of warming-induced abiotic and biotic factors on microbial physiology, as responded in A2.

We agree with the Reviewer that warming can affect microbial physiology via changes in soil moisture which was measured using field sensors for the duration of the experiment, as we describe in our Method (see Lines 573-576). We respectfully refer the Reviewer to our previous responses (A2) to their comments about the need for greater reference to the importance of management on soil moisture.

We also thank the Reviewer for bringing this useful reference to our attention. We have included it in the Introduction and Discussion sections (Lines 114-116; Lines 458-461).

- Lines 573-576 “Soil temperature at 5 cm depth was monitored with PT 100 thermocouple, while volumetric soil moisture at 0-10 cm depth was measured by FDS100 soil moisture sensors (Unism Technologies Incorporated, Beijing).”
- Lines 114-116 “Generally, warming decreases microbial CUE, as a greater proportion of the substrate is reallocated from growth to maintenance metabolism^{19,27}, which alters rates of enzyme-driven processes²⁸⁻³⁰.”
- Lines 459-461 “This is also consistent with the observation that microorganisms may downregulate their protein production machinery in response to warming-induced substrate depletion³⁰.”

A20. L135-138: *Is this experiment also a warming experiment? From this description we can't understand if it is a warming and tillage experiment. Please rephrase this section to clarify. In the last paragraph of the introduction it is important that authors concisely summarize their methods so that authors can understand their results section.*

Response: We apologize to the Reviewer for the lack of clarity. We have revised the last paragraph of the Introduction section to improve a more concise description of the methods employed in our study:

Lines 139-141 “Herein, we present the first study from a long-term agricultural experiment that that spans a decade and encompasses two distinct management systems (conservation versus conventional agriculture) × two warming levels (warming versus ambient).”

A21. L812: *five or three times?*

Response: Sorry for this mistake. It should be “three times”. We have corrected this in the revised text:

Lines 697-699 “Finally, the exudates were collected by flushing the cuvette three times with 15 ml of the fresh nutrient solution after the 24-h incubation.”

A22. L384: *replace was by is required.*

Response: We thank the Reviewer for the comment. This sentence has been deleted in the revised text.

A23. Data availability statement: authors could make available in the supplementary files some files with their raw dataset. Right now authors write “The authors declare that the data supporting the findings of this study are availability with the article and its supplementary Information files, and from the corresponding authors on request.” However, none of their supplementary files contain their raw data.

Response: We appreciate the Reviewer for their advice. We have provided the source data in the revised text.

B. Responses to Reviewer #2 (Remarks to the Author):

B1. *The article “Microbially mediated mechanisms underlie soil carbon accrual by conservation agriculture under decade-long warming” provides results from a multi-factor field experiment including warming and management treatments. The primary focus of the paper was conventional tillage and its role in accrual of soil organic carbon over the experiment. The authors present evidence of microbial community shifts, particularly the fungal community, that may be responsible for the increase in SOC. The heavy focus on microbial measures allows for a more mechanistic understanding of why SOC accrual is occurring under conservation tillage even with warming. This helps improve the understanding of SOC formation and stabilization in soils, and highlights the role of fungi in this accumulation.*

The methodology used in this manuscript are appropriate. In particular, the use of isotopes for CUE improves the accuracy of the CUE estimates. The results support the interpretation and conclusion of the work. Figure 6 is very helpful for following the conclusions.

Response: We are grateful to the Reviewer for their positive and supportive comments and for highlighting the strengths and novelty of our study.

B2. *One suggestion in the results: The results text initially follows a pattern where conservation tillage data is presented and then traditional tillage. Then it switches to traditional followed by conservation. It might easier for the reader to follow if the same order was maintained throughout the results.*

Response: We thank the Reviewer for their advice on the presentation of our Results. We have corrected the Results section text to keep them consistent.

C. Responses to Reviewer #3 (Remarks to the Author):

C1. *The authors studied the effect of warming in combination with ‘conservation agriculture’ on Soil Organic Carbon (SOC) dynamics in an agricultural soil in northern China. They found that accelerated fungal succession and microbial community-level physiological adjustment towards high growth efficiencies and metabolic byproducts under conservation agriculture promoted SOC formation and accrual in warmed conditions over 10 years.*

The topic of this research is important and of great interest to the readers. As global climate change is rapidly advancing, and the earth is warming, we will increasingly require agricultural practices to adapt so that more carbon can be stored in soils. I also find that the experimental design of the experiment is sound and that the manuscript contains extensive amounts of data, including measurements of dissolved organic carbon, root biomass and root exudation carbon, aboveground biomass indexes, phospholipid fatty acid analysis, bacterial and fungal rRNA gene amplicon sequencing, glucosamine, galactosamine and muramic acid analysis to characterize necromass, metagenomic sequencing, and microbial carbon use efficiency measurements using the ^{18}O -H $_2\text{O}$ tracer method, which also provides measures of microbial growth. This is a rich and important data set.

Response: We are grateful to the Reviewer for their positive and supportive comments and for highlighting the strengths of our approach.

C2. *However, I don’t find the manuscript to be ready for publication because the writing is not precise and it is often unclear what argument the authors are making. Let me illustrate this by considering the abstract of the paper. I focus only on the abstract because it is an important component of a research paper that should summarize the findings of the paper and because I don’t have the time nor resources to rewrite the entire paper. But the entire paper not just the abstract needs to be rewritten.*

Response: We thank the Reviewer for their suggestions to improve our manuscript. We have revised the Abstract section and the manuscript thoroughly to be clearer. We hope that the changes are appropriate and meet the standards for publication.

C3. *L53 Line 53 mentions “conservation agriculture.” Define more clearly what conservation agriculture is. Is it simply “no tillage”? If so why not call it that?*

Response: We thank the Reviewer for their advice. Conservation agriculture represents a suite of crop management principles, of which reduced or zero/no tillage and permanent soil cover by crop residues (or a cover crop) are part, and that were applied in our long-term experiment. We have added the definitions of the two management treatments to the Introduction section to provide clarity (Lines 98-99; Lines 148-151).

To avoid the misunderstanding, and also to make it clearer, we use the term conservation agriculture for the no-till treatment, considering that the 2 other requirements are also fulfilled. We have replaced traditional tillage with conventional agriculture. To also address the other Reviewer’s request for consistent use of terms, we have replaced conventional agriculture with traditional tillage in the manuscript and figure legends throughout the manuscript.

- Lines 97-99 “Soils managed under conservation agriculture, which involve increased retention of organic residue and reduced or zero tillage¹⁴, should contain more SOC.”
- Lines 144-147 “Specifically, our study aimed to: (i) assess whether warming differentially affects SOC accrual under conservation agriculture (chopped crop residues returned and no tillage) versus conventional agriculture (crop residue removed and annual tillage).”

C4. L53 “A positive warming effect on SOC strengthened under conservation agriculture” It is unclear what a positive warming effect is. Does this mean that warming increased SOC concentrations?

Response: We thank the Reviewer for pointing out this lack of clarity. We have revised this sentence to make it clearer; it now reads:

Lines 52-55 “Our results revealed that warming increased SOC content and accelerated fungal community temporal turnover under conservation agriculture, but not under conventional agriculture.”

C5. L54 Warming accelerated fungal community turnover under conservation agriculture, as indicated by high temporal scaling rates and increasingly divergent succession.” What are high temporal scaling rates and divergent succession? Does this mean there was an increase in fungal diversity under the warming treatment?

Response: We apologize to the Reviewer for the lack of clarity. We investigated microbial community turnover by using time-decay relationships (TDR). The slopes of the TDRs’ values reflect the temporal turnover rates of soil microbial communities. We also calculated the paired community difference between warming and no-warming under the two management types in this study. The increased paired community difference between warming and no-warming reflects the divergent succession. We revised this sentence to make it clearer (Lines 58-59); it now reads:

Lines 52-55 “Our results revealed that warming increased SOC content and accelerated fungal community temporal turnover under conservation agriculture, but not under conventional agriculture.”

C6. L59 “In parallel, warming increased microbial growth-related genes”. It is unclear to me how warming can increase genes. Does this mean that the relative abundance of growth related genes increased? Or is it absolute abundance? Or did the diversity of growth related genes increase?

Response: We apologize to the Reviewer for this imprecise expression, since we refer here to relative abundance. Given the Abstract word length, we have deleted this statement from the Abstract. However, we did check the correct use of the expression in the main manuscript.

C7. L63 microbial community-level physiological adjustment towards high growth efficiencies and metabolic byproducts”. What is a community level physiological adjustment? And how can it be adjusted towards metabolic by products?

Response: We thank the Reviewer for pointing out the lack of clarity. In this study, we measured the CUE by using ¹⁸O-H₂O. This parameter defines community-scale microbial efficiency as gross biomass production per unit substrate taken up over short time scales (Geyer et al., 2016). Therefore, we used the expression “community-level physiological adjustment”, but we see that it could generate confusion, and have removed it from the revised text. We also have checked the use of the expression in the main manuscript.

Geyer, K. M., E. Kyker-Snowman, A. S. Grandy and S. D. Frey (2016). Microbial carbon use efficiency: accounting for population, community, and ecosystem-scale controls over the fate of metabolized organic matter. *Biogeochemistry* 127(2-3): 173-188.

C8. In my opinion before the manuscript can be properly reviewed it is important that the writing becomes more precise so that its meaning is unambiguous and not left to the interpretation of the reader.

Response: We appreciate the Reviewer for their comments and patience. We have thoroughly reviewed and revised the whole manuscript to avoid ambiguity. We hope that the quality and readability are substantially improved and that concepts are more precisely described to the Reviewer's satisfaction.

REVIEWER COMMENTS

Reviewer #1 (Remarks to the Author):

Tian et al. improved the manuscript considerably after the first submission. I appreciate that the authors have carefully implemented the reviewers' comments/suggestions. In my opinion, the paper reads very well, but the methods section needs more clarification, as I explain below.

I have one major question: in the methods section authors state "Soil samples (0-5 cm depth) were taken using a soil corer (5 cm inner diameter) after winter wheat harvest in 2010, 2012, 2014, 2016, 2018 and 2020. Analysis for ... microbial community CUE, bacterial diversity and fungal diversity, were conducted in 2010, 2012, 2014, 2014, 2018 and 2020 on the samples collected in those years." Authors mentioned that they used the 18O water method for the CUE and they cite the studies of Spohn et al., 2016 and Zheng et al., 2019. How did authors perform the CUE measurements using this method in 2010, 2012 and 2014? Those samplings occurred before the publication of the method (first published in 2016). Authors need to clarify this.

Figure SEM: I would like to ask authors to verify their SEM output. While their results suggest that warming had a positive impact on substrate availability their SEM show the opposite. The same contradictory result is shown for moisture. One suggestion would be to show 2 distinct models: one model under conservation agriculture and one model for the conventional agriculture. Like this authors could probably visualize better how the distinct agricultural systems are modifying the mechanisms underlying SOC changes. Right now their SEM is showing contradictory results with some of the other figures which is a bit worrying. Please also report the number of samples used to run the SEM.

L265-268: Your hypothesis doesn't really match this statement. In L155-157 you state your hypothesis II: "(ii) microbial community-level adaptation to warming and higher microbial growth efficiency and metabolic functions in response to larger substrate availability increases the contribution of microbial necromass to SOC over time under conservation agriculture;" Their results don't allow to evaluate microbial physiological response to warming exclusively without considering the higher input in plant biomass. Please rephrase this sentence.

L300-303: temporal turnover of microbial community structure...

L318: community turnover

L420-421: "warming increased aboveground and belowground biomass and root exudation, regardless of management systems (Figs. 1d-1g)" This statement doesn't correspond to the results. Please correct it.

L427-429: the SEM results are contradictory to this statement. Please consider modifying the SEM as suggested previously or discuss here why the SEM is suggesting a different outcome.

Methods section: the methods section is still lacking in clarity and requires more information to allow the readers to understand their methods.

One very important aspect of this study is that authors argue that they collected data and are presenting results compiling an interval of 10 years (2010-2020). The way authors describe their methods suggests that they always used the same methods during this 10 years interval. Did authors always used an Illumina NovaSeq 6000 to sequence their soils? Their method section lead us to this conclusion but was this technology available in 2010? I don't think so. Was the soil frozen and all samples sequenced at the same time? This is not what is written in the methods. Please clarify your methods section.

Regarding the CUE measurements authors should also provide the info of which laboratory they performed the IRMS analysis.

Reviewer #3 (Remarks to the Author):

The authors have done a nice job addressing my concerns and I think the manuscript is suitable for publication.

Response to Reviewers' comments

A. Response to Reviewer #1 (Remarks to the Author):

A1: Tian et al. improved the manuscript considerably after the first submission. I appreciate that the authors have carefully implemented the reviewers' comments/suggestions. In my opinion, the paper reads very well, but the methods section needs more clarification, as I explain below.

Response: We thank the Reviewer for supportive comments and further helpful recommendations to improve our manuscript.

A2: I have one major question: in the methods section authors state "Soil samples (0-5 cm depth) were taken using a soil corer (5 cm inner diameter) after winter wheat harvest in 2010, 2012, 2014, 2016, 2018 and 2020. Analysis for ... microbial community CUE, bacterial diversity and fungal diversity, were conducted in 2010, 2012, 2014, 2014, 2018 and 2020 on the samples collected in those years." Authors mentioned that they used the 18O water method for the CUE and they cite the studies of Spohn et al., 2016 and Zheng et al., 2019. How did authors perform the CUE measurements using this method in 2010, 2012 and 2014? Those samplings occurred before the publication of the method (first published in 2016). Authors need to clarify this.

Response: We apologize for the lack of clarity in the 'Field measurements, soil sampling and analyses' part of the Methods section in the previous version. The soil samples were frozen in the respective years, and analyzed for ¹⁸O-CUE in 2021. We have revised the text to improve clarity (please see lines 572-601).

In this long-term field experiment, the soil samples were collected every 2 years after the winter wheat harvest from 2010 to 2020. Sub-samples of the initial sample were stored in a freezer at -80 °C after sampling. On the specific point of the application of the Spohn et al.: Schroeder et al. (2021) showed that CUE was unaffected by soil sample pre-treatment, including freezing at -80 °C. This finding presented us with an opportunity to apply the method of Spohn et al. to soil samples that were frozen from 2010 to 2020. Therefore, we analyzed the CUE by ¹⁸O-H₂O approach after 7 days of pre-incubation of the soil stored at -80 °C in 2021.

We have added the information in the revised text: Microbial CUE was measured in 2021 in all soils (sampled in 2010, 2012, 2014, 2016, 2018, and 2020) that had been stored at -80°C using the ¹⁸O-H₂O tracer method after 7 days preincubation^{54,55} (details below) (Lines 591-593). We have also referred to Schroeder et al. in the 'Microbial carbon use efficiency' part of the Methods section at Line 666.

Schroeder, J., Kammann, L., Helfrich, M., Tebbe, C.C. and Poeplau, C., 2021. Impact of common sample pre-treatments on key soil microbial properties. *Soil Biology and Biochemistry*, 160, p.108321.

A3: Figure SEM: I would like to ask authors to verify their SEM output. While their results suggest that warming had a positive impact on substrate availability their SEM show the opposite. The same contradictory result is shown for moisture. One suggestion would be to show 2 distinct models: one model under conservation agriculture and one model for the conventional agriculture. Like this authors could probably visualize better how the distinct agricultural systems are modifying the

mechanisms underlying SOC changes. Right now their SEM is showing contradictory results with some of the other figures which is a bit worrying. Please also report the number of samples used to run the SEM.

Response: We appreciate the Reviewer's inquiry and recommendation which led us to thoroughly check our data and SEM analysis.

The negative direct effect of temperature on substrate availability was compensated by the positive effects of moisture due to the strong collinearity between temperature and moisture ($R^2 > 0.78$): Temperature increase has led to a decrease in soil moisture. Therefore, the total effect of temperature on substrate availability will be the following: an increase in temperature increases substrate availability because of a decrease of moisture with increasing temperature. Because of this, only temperature was selected as the representative variable for a better model fit in the revised text. After modifying the model based on the Reviewer's suggestion, the results were clearer and the effect of temperature was emphasized. Please See Fig.5. The number of samples used to run the SEM is also provided at Line 795-796 and in the Fig.5.

We also tried to analyze the data with two models for the two systems based on the Reviewer's suggestions, but the model fit was poor for the conventional agriculture treatment. Therefore, we prefer to combining them. We hope the Reviewer finds this acceptable.

A4: L265-268: Your hypothesis doesn't really match this statement. In L155-157 you state your hypothesis II: "(ii) microbial community-level adaptation to warming and higher microbial growth efficiency and metabolic functions in response to larger substrate availability increases the contribution of microbial necromass to SOC over time under conservation agriculture;" Their results don't allow to evaluate microbial physiological response to warming exclusively without considering the higher input in plant biomass. Please rephrase this sentence.

Response: We appreciate the Reviewer's helpful comments. We have modified the text accordingly:

"The increase in microbial growth efficiency and total necromass under conservation agriculture in warmed soils, in parallel with larger plant carbon input and increasing SOC content, suggests that the microbial community and its physiological responses adapted to warming conditions over time, which supports the hypothesis (ii)". (Lines 265-269).

A5: L300-303: temporal turnover of microbial community structure...

Response: We thank the Reviewer for pointing this out. We have corrected this in the revised text.

"To elucidate the impacts of warming on the temporal turnover of microbial community structure, we assessed the time-decay relationships (TDRs) for bacteria and fungi." (Lines 301-303).

A6: L318: community turnover

Response: We thank the Reviewer for pointing this out. We have corrected this in the revised text.

A7: L420-421: “warming increased aboveground and belowground biomass and root exudation, regardless of management systems (Figs. 1d-1g)” This statement doesn’t correspond to the results. Please correct it.

Response: We appreciate the Reviewer’s comments. We have improved the revised text (Lines 419-421) which now reads ‘Though warming increased belowground biomass and root exudation (Figs. 1e-1f), these changes were not translated to an increase in SOC under warmed conventional agriculture.’

A8: L427-429: the SEM results are contradictory to this statement. Please consider modifying the SEM as suggested previously or discuss here why the SEM is suggesting a different outcome.

Response: We appreciate the Reviewer’s comments. We have modified the SEM according to the Reviewer’s suggestions. Please see our response to comment A3.

A9: Methods section: the methods section is still lacking in clarity and requires more information to allow the readers to understand their methods.

One very important aspect of this study is that authors argue that they collected data and are presenting results compiling an interval of 10 years (2010-2020). The way authors describe their methods suggests that they always used the same methods during this 10 years interval. Did authors always use an Illumina NovaSeq 6000 to sequence their soils? Their method section lead us to this conclusion but was this technology available in 2010? I don’t think so. Was the soil frozen and all samples sequenced at the same time? This is not what is written in the methods. Please clarify your methods section.

Response: We apologize for the lack of clarity in the ‘Field measurements, soil sampling and analyses’ part of the Methods section. Specifically, regarding the timing of the sequencing analysis: Sub-samples of soil were stored at -80 °C after each sampling occasion in 2010, 2012, 2014, 2014, 2018, and 2020. DNA was extracted and analyzed for bacterial and fungal diversity from those frozen samples at the same time in 2021. We have added this information in the revised text (Lines 593-596). Please also see our response to question A2. We hope that the improved Methods section is now clear and acceptable.

A10: Regarding the CUE measurements authors should also provide the info of which laboratory they performed the IRMS analysis.

Response: We apologize to the Reviewer for the lack of clarity. We have added the information in the revised text at Lines 677-683, which now reads: ‘The extracted DNA (50 µL) was dried in silver capsules at 60 °C for 2 days. Subsequently, the ¹⁸O abundance and total O content were determined using an elemental analyzer (FLASH 2000, Thermo Fisher Scientific, Cambridge, UK) coupled with an isotope ratio mass spectrometer (IRMS) system (ConFlo VI interface and MAT253 IRMS, Thermo Scientific, Bremen, Germany) at the Chinese Academy of Sciences Institute of Subtropical Agriculture, (Changsha, Hunan Province, China).’

B. Response to Reviewer #3 (Remarks to the Author):

B1: The authors have done a nice job addressing my concerns and I think the manuscript is suitable for publication.

Response: We appreciate the Reviewer's supportive comments and recommendation for the suitability of our manuscript for publication.

REVIEWERS' COMMENTS

Editor's note: the reviewers were satisfied with the revisions and had no further comments